# Haloturbation in the northern Atacama Desert revealed by a hidden subsurface network of calcium sulphate wedges

Aline Zinelabedin[1,2], Joel Mohren[1,3], Maria Wierzbicka-Wieczorek[1], Tibor J. Dunai[1], Stefan Heinze[4] Benedikt Ritter[1]

[1]Institute of Geology and Mineralogy, University of Cologne, Zülpicher Str. 49b, 50674 Cologne, Germany
[2]Institute of Geography, University of Cologne, Zülpicher Str. 45, 50674 Cologne, Germany
[3]Department of Geography, RWTH Aachen University, Wüllnerstr. 5b, 52062 Aachen, Germany
[4]Institute for Nuclear Physics, University of Cologne, Zülpicher Str. 77, 50937 Cologne, Germany

*Correspondence to*: Aline Zinelabedin (aline.zinelabedin@uni-koeln.de)

**Abstract.** While the formation of periglacial wedges and polygonal patterned grounds has been extensively studied and many of the processes involved have been understood, knowledge on the formation of similar features found in arid to hyperarid environments remains largely rudimentary. This study aims to fill the existing knowledge gap by examining a network of vertically laminated, calcium sulphate-rich wedges that extend to depths of 1.5–2.0 meters in the alluvial subsurface of the Aroma fan in the northern Atacama Desert. The subsurface wedges are characterised by their high anhydrite content, distinguishing them from the wedges and polygon structures found at other sites in the Atacama Desert. These structures appear to have been predominantly formed by thermal contraction or desiccation processes in playa-like environments. In contrast, it is hypothesised that haloturbation mechanisms, specifically the swelling and shrinking due to the hydration and dehydration of calcium sulphate, are the primary factors driving wedge formation at the Aroma fan site. Haloturbation processes require the input of moisture, and Aroma fan wedge formation is therefore likely to be associated with meteoric water received from sporadic rain events during predominantly arid to hyperarid climates. The subsurface wedge network is covered by a stratigraphically younger surface crust primarily composed of gypsum. The presence of the surface crust may indicate a shift towards drier environmental conditions, which enabled the accumulation and surface inflation of calcium sulphate and other salts through atmospheric deposition. A climatic shift could have resulted in a deceleration of haloturbation processes in the subsurface. However, modern sediment transport from the surface into the subsurface still appears to occur along cracks within the crust. To gain a thorough understanding of the complex mechanisms and rates involved in wedge formation, it is crucial to establish a geochronological framework based directly on wedge and crust material. The temporal resolution of wedge growth stored within the sequence of vertical laminae offer the potential for the calcium sulphate wedges to be used as palaeoclimate archives, which could contribute to the understanding of wedge and polygonal patterned ground formation in other water-limited environments, such as Mars.

## 1 Introduction

Geomorphological features such as subsurface wedges and polygonal patterned grounds are commonly found in periglacial environments (e.g. Lachenbruch, 1962; Washburn, 1956, 1979; Black 1976; Mackay, 1990). These features have been successfully used as paleoclimate and paleoenvironmental archives (e.g. Williams, 1986; Liu and Lai, 2013; Opel et al. 2018; Campbell-Heaton et al., 2021). In general, wedge and polygon formation under periglacial conditions is driven by thermal contraction mechanisms (Edelman et al., 1936; Lachenbruch, 1962). The formation of ice wedges is primarily influenced by the behaviour of water in both its liquid and solid forms. The laminated structure of periglacial ice wedge results from the freeze-thaw cycles of ground ice (Edelman et al., 1936; Lachenbruch, 1962). Polygonal patterned grounds are classified into two types: in sorted and non-sorted polygons. The differentiation between these two categories is determined by the type of subsurface deformation (Washburn, 1956). The formation of sorted polygons is driven by freeze-thaw cycles in the active layer and frost heave processes in polar and high alpine environments. These processes result in sediment being sorted by grain size, ultimately leading to the formation of sorted stone circles (Kessler and Werner, 2003). In contrast, non-sorted polygonal patterned grounds in arctic to subarctic environments are defined by cracks and indicate the presence of subsurface wedge structures. The formation of non-sorted polygons can vary depending on the material and environmental conditions (Certini and Ugolini, 2015). The presence of analogous polygonal patterns on the Martian surface has been observed by satellite imagery and correlated with the periglacial environment on Earth. This observation has profound implications for the interpretation of geomorphology, surface processes, and water availability on Mars (e.g. Levy et al., 2010; Hauber et al., 2011; Soare et al., 2014; Amundson, 2018).

In comparison to a water-rich periglacial environment, strongly differing environmental conditions prevail in the arid to hyperarid Atacama Desert, where landscape-modifying processes are influenced by severe water scarcity. However, polygonal patterned grounds associated with subsurface wedge structures that are strikingly similar to periglacial ice wedge structures can also be found in the Atacama Desert (Ericksen, 1981, 1983; Allmendinger and González, 2010; Buck et al., 2006; Howell et al., 2006; Rech, et al., 2006; Howell 2009; Rech et al. 2019; Pfeiffer et al., 2021; Sager et al., 2021, 2022; Zinelabedin et al., 2022). In contrast to periglacial environments, the formation of polygonal patterned grounds and wedge structures in the Atacama Desert is not controlled and dominated by the interaction of ice and liquid water. This suggests that other processes are responsible for wedge-polygon-formation under hyperarid conditions. Several studies have proposed different formation processes for wedge structures and polygonal patterned grounds formed in such arid environments. These include haloturbation processes (e.g. Chatterji and Jeffrey, 1963; Buck et al. 2006; Ewing et al., 2006; Howell et al. 2006; Howell, 2009; Zinelabedin et al., 2022), thermal contraction (Yungay region; see Sager et al., 2021), and desiccation processes in playa-like environments (e.g. Atacama Desert; Ericksen, 1981, 1983; N-America;

Neal et al. 1968). Thermal contraction processes occur as a consequence of tensile stresses that develop in deposits during cooling (Lachenbruch, 1962). Desiccation cracking is, among other factors, caused by the dehydration of deposits due to the phase transition of hydrous calcium sulphate (gypsum; $CaSO_4 \cdot 2H_2O$) to anhydrous calcium sulphate (anhydrite; $CaSO_4$) (Cooke and Warren, 1973; Tucker, 1981).

The term "haloturbation" is typically used to describe all salt-related processes that modify the original structure of host deposits (rocks, sediments, soils). The presence of evaporites is a crucial factor in determining whether haloturbation is occurring. However, the definition of this term varies inconsistently in the literature. Some studies describe haloturbation as involving dissolution and reprecipitation (e.g., Rychliński et al., 2014), while others include swelling and shrinking due to the hydration and dehydration of salts (e.g., May et al., 2019).

We specifically use the term haloturbation to emphasize that salts are the primary agents and limiting factors in the processes encompassed by haloturbation (dissolution, reprecipitation, shrinking/dehydration, swelling/hydration). In this study, we focus on calcium sulphate-related haloturbation processes, as calcium sulphate is a dominant component of the surface deposits in the hyperarid Atacama Desert (Ericksen, 1983; Rech et al., 2003; Ewing et al., 2006; Wang et al., 2014).

One of the primary mechanisms of haloturbation is the dissolution and subsequent (re-)precipitation of salt minerals within pore spaces, which can result in salt heave and the rearrangement of deposits (e.g. Tucker, 1981; Fookes and Lee, 2018). This process can also lead to the fracturing of clasts. The direct precipitation of anhydrite from a solution results in the formation of the calcium sulphate polymorph β-anhydrite (β-$CaSO_4$; insoluble), which is thermodynamically stable under the ambient conditions prevailing in the Atacama Desert

(Tang et al., 2019; Beaugnon et al., 2020). As a consequence, this anhydrite polymorph occurs naturally in evaporite deposits (Beaugnon et al., 2020).

The second mechanism, which we summarise under the general term haloturbation, is characterised by swelling and shrinking caused by phase transitions due to hydration and dehydration of calcium sulphates. The phase transition of the thermodynamically metastable and soluble γ-anhydrite (γ-$CaSO_4$; Tang et al., 2019; Beaugnon et

al., 2020) over bassanite (hemihydrate; $CaSO_4 \cdot 0.5H_2O$) to gypsum ($CaSO_4 \cdot 2H_2O$), is accompanied by a volume increase of ~61% (Butscher et al., 2017, 2018; Jarzyna et al. 2021). The reversal process results in a volume decrease of ~29% for the gypsum-bassanite transition (Milsch et al., 2011), and ~39% of the total volume decrease from gypsum to γ-anhydrite in an open system (Milsch et al., 2011; Sanzeni et al., 2016).

The occurrence of salt-related processes, including swelling (hydration) and shrinking (dehydration), dissolution

and precipitation in salt-bearing deposits within the hyperarid core of the Atacama Desert is linked to the persistence of hyperarid conditions since at least the Early Miocene (Dunai et al., 2005; Evenstar et al. 2009; Jordan et al., 2014; Evenstar et al., 2017; Ritter et al., 2018, Ritter et al. 2022). Sporadic precipitation events have been observed to result in considerably low rates of erosion (Kober et al., 2007; Placzek et al., 2010, 2014; Starke et al., 2017; Mohren et al., 2020, Ritter et al. 2023), which in turn has led to the long-term preservation of

surfaces (Dunai et al., 2005; Nishiizumi et al., 2005; Kober et al., 2007, Evenstar, et al., 2017; Ritter et al. 2019, 2022) and the accumulation of landscape-draping calcium sulphate-rich soils by atmospheric deposition (Ericksen, 1981, 1983; Rech et al., 2003; Michalski et al., 2004; Ewing et al., 2006; Wang et al., 2014, 2015; Rech et al., 2019). Nonetheless, hyperaridity in the Atacama Desert is repeatedly interrupted by wetter but still (hyper-)arid conditions (e.g. Dunai et al. 2005; Jordan et al. 2014; Evenstar et al. 2017; Ritter et al., 2018, 2019, Diederich et al. 2020; Medialdea et al. 2020; Ritter et al. 2022, Wennrich et al., 2024). These episodes appear to provide sufficient moisture to 'activate' salt dynamics in evaporite-bearing deposits (e.g. Buck et al., 2006; Howell et al., 2006; Howell, 2009; Wang et al., 2015; Rech et al. 2019).

The Atacama Desert's moisture input comes from two primary sources: coastal fog (span. 'camanchaca'; see Cereceda et al. 2008) and sporadic rain events (recent precipitation events, e.g. described in Bozkurt et al. 2016, Vicencio Veloso 2022, Cabré et al. 2022, Wennrich et al., 2024). The formation of coastal fog is attributed to a persistent atmospheric inversion layer, which traps moist Pacific air below ~1000 meters a.s.l. (Houston, 2006; Cereceda et al. 2008; Garreaud et al. 2009; Schween et al. 2020). The maximum altitude of present-day fog is ~1200 m a.s.l. (based on the fog-dependent spatial distribution of *Tillandsia landbeckii* sp.; Cereceda et al. 2008). The eastward movement of the coastal fog is limited by the high coastal cliffs, but it can cross this barrier by traveling through deep canyons (span. 'quebradas'; e.g. Río Loa, Tiliviche canyon). In an environment characterised by extreme water scarcity, the fog is considered to be one of the primary agents for surface activity and modification in the hyperarid core of the Atacama Desert (Ericksen, 1981,1983; Rech et al., 2003). The probability of fog migration towards the hinterland decreases in eastern direction (del Río et al., 2018), such that the Central Depression or Precordillera receive little – if any – fog moisture. However, the atmospheric inversion layer has been observed to undergone vertical displacement on both modern (del Río et al., 2018, Muñoz et al., 2016, Schween et al., 2020, Böhm et al., 2021) and Holocene (Latorre et al., 2011) timescales. These findings may be used to support the hypothesis that a consistent supply of remote inland surfaces with fog moisture could be achieved in the long term. Infrequent rain events that occasionally approach the inner Atacama Desert (e.g. Jordan et al., 2019) are typically caused by cut-off low pressure systems (Reyers et al., 2021). These systems may enter the Atacama Desert either from the north (see Böhm et al. 2021) or from the south (Stuut and Lamy, 2004).

While these environmental conditions seem favourable for the formation of polygonal patterned grounds and wedge structures, the specific agents and processes responsible for wedge polygon formation in arid to hyperarid environments are still the focus of ongoing research. Moreover, the timescales and environmental conditions under which the wedges form remain unresolved. Understanding the processes behind the formation of hyperarid wedges could enable the interpretation of laminated wedge structures as records of past environmental conditions. The identification of wedge-forming processes in (hyper-)arid environments suggests that similar processes may also contribute to the formation of extra-terrestrial geomorphological features, such as those

observed on Mars (e.g. Sager et al., 2021; Sager et al., 2023). Amundson (2018) suggests that the weathered soil from the surface of the Meridiani Planum on Mars is comparable to the chemistry and morphology of hyperarid soils on Earth. Previous studies have also documented the occurrence and distribution of hydrous sulphates on the Martian surface (Gendrin et el., 2005) and Ca-Mg sulphates on polygon ridges in the Gale Crater on Mars (Rapin et al., 2023). Therefore, a comparison of (hyper-)arid polygonal patterned soils with extra-terrestrial counterparts bears the potential to provide important insights for future Mars exploration.

The present study aims at resolving the processes and mechanisms governing wedge formation in a hyperarid setting, using calcium sulphate wedges formed in coarse-grained deposits of the Aroma alluvial fan situated on the Andean foreslope as study subjects. We propose different formation scenarios of subsurface wedge and surface crust development resulting from local haloturbation processes. These scenarios are supported by an extensive set of geochemical, mineralogical, sedimentological, and microscopic studies. In this study, we differentiate between the Aroma wedge structures and wedge-polygon-formation processes observed in other regions of the Atacama Desert. These encompass thermal contraction mechanisms in the Yungay region (Sager et al., 2021) and desiccation polygon structures in playa-like environments (e.g., Ericksen 1981, 1983; Bobst et al., 2001; Finstad et al., 2016).

## 2 Regional setting

The study site is located on the western Andean foreslope, north of the Quebrada Aroma (19° 39' 34.02''S, 69° 35' 51.4''W; Fig. 1A, B). On a regional scale, the western Andean foreslope is bordered by the rising Western Cordillera to the east and by the Central Depression and Coastal Cordillera to the west (Fig. 1A). Climatic conditions in the study area are characterised as hyperarid with a mean annual precipitation of <2 mm yr$^{-1}$ (see isohyets in Fig. 1A; Houston, 2006). The Aroma fan and adjacent alluvial fans consist of alluvial gravels affiliated to the Upper El Diablo Formation of Middle to Late Miocene age (Muñoz and Sepúlveda, 1992; Farías et al., 2005; von Rotz et al., 2005; Evenstar et al., 2009; Hartley and Evenstar, 2010; Lehmann, 2013; Cosentino and Jordan, 2017), covered by gypsic relict soils and gypsisols (Cosentino and Jordan, 2017). In the vicinity of the study area, the Aroma fan surface has a mean slope of about 1.5° (Evenstar et al., 2009). At an altitude of ~1630 m a.s.l., we found a ~20–30 m long trench located adjacent to the A-457 road which had been excavated presumably during road construction works prior to 2017. The excavation exposed a network of subsurface soil cracks, which are up to ~1–2 m deep, and vertically laminated wedges developed within the alluvium of the Upper El Diablo Formation (Fig. 1C, D, E, G). Shattered cobbles and boulders (parent material) infiltrated by anhydrite and embedded in a calcium sulphate-rich matrix appear in the host sediment between the wedge structures (Fig 1H, I). The wedge network is covered by a ~20 cm thick surface crust dominated by gypsum (Fig. 1E, F) which does not appear to form a polygonal patterned ground at its surface (Fig. 1B.1).

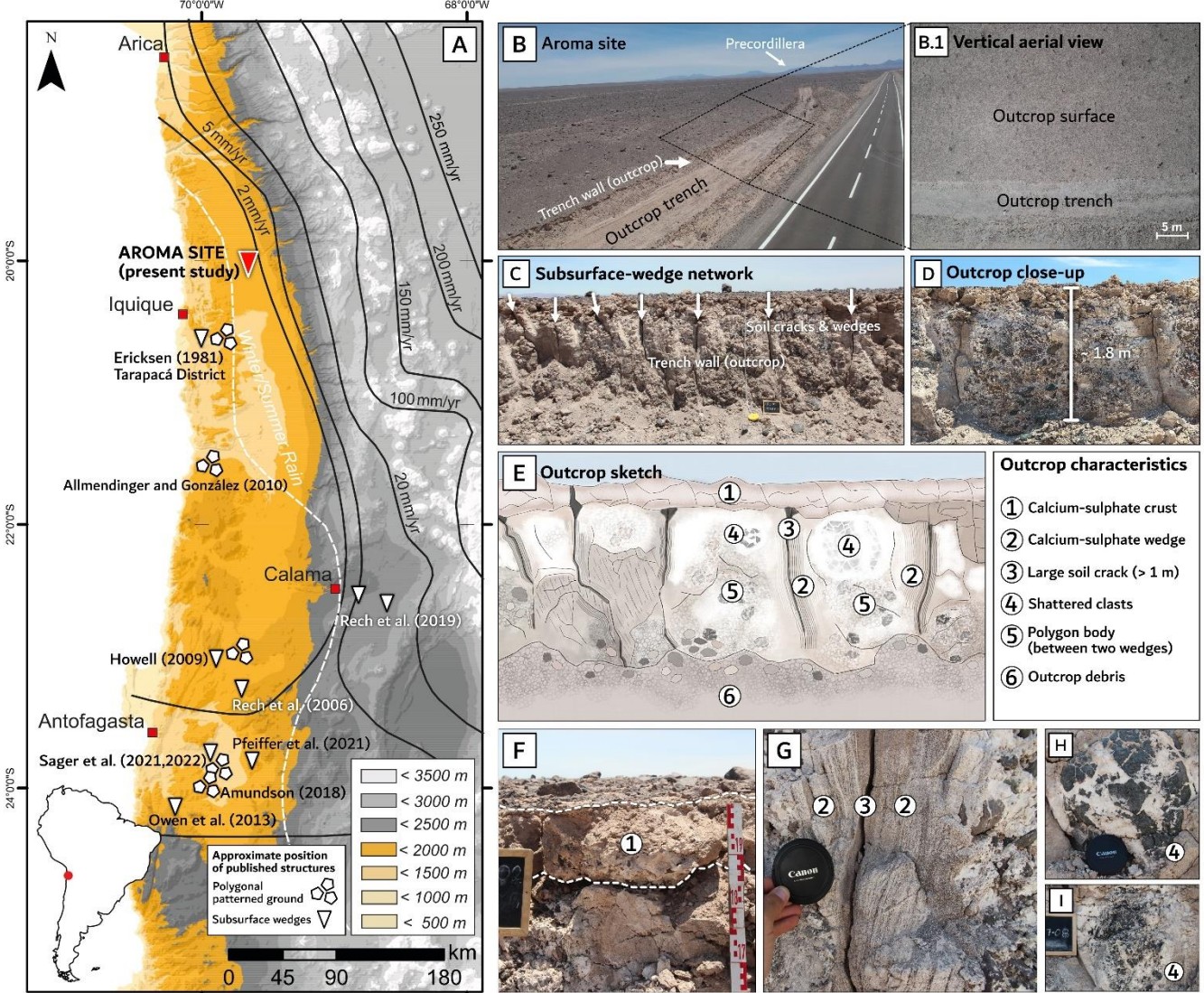

**Figure 1: A)** Colour-shaded digital elevation model (derived from SRTM-data, created using ArcMap ver. 10.5.1) of northern Chile including isohyets (modified from Houston, 2006) as well as winter-rain and summer-rain dominated areas (white dashed line) after Houston (2006). The red point in the South America map in the lower left corner displays the position of the study site. The red inverted triangle in the map of northern Chile indicates the study site situated on the Aroma fan. White inverted triangles and white polygons display published studies which investigated subsurface wedge structures and surface polygonal patterned grounds in the Atacama Desert. The red squares represent larger cities. **B)** Oblique drone image of the Aroma outcrop viewing in NE direction. **B.1)** Nadir view of the outcrop surface showing no indications of polygonal patterned ground or similar surface expression. **C)** Outcrop trench wall (viewing in NW direction) displaying a subsurface network of vertically laminated wedge structures with vertical soil cracks (> 1 m) along their centres. **D)** Close-up of outcrop structures and scale (viewing in NW direction). **E)** Outcrop sketch highlighting all important characteristics of the outcrop trench wall. **F)** Close-up photograph of the ~20 cm thick surface crust. **G)** Close-up photograph of subsurface crack and subsurface wedge parts to the left and to the right side of a soil crack. **H)** Shattered clast damaged by calcium sulphate intrusion. **I)** Shattered clast with a higher degree of destruction as in photograph H.

## 3 Material and methods

Sampling was conducted during two field campaigns in September 2017 and October 2018; all samples were collected from the northwestern trench wall. The sample set consists of six surface quartz clasts of pebble to cobble size sampled for surface exposure dating (see Fig. S.5 in the supplementary material), a surface crust
block (0 cm to ~18–20 cm depth) containing various vertical and horizontal cracks, two subsurface wedge parts from ~40–50 cm depth, and two shattered clasts from the polygon body from ~70–80 cm (ARO18-05) and ~100–110 cm (ARO18-04) depth (unconsolidated alluvium of the Upper El Diablo Formation). Figure 2 shows a schematic profile of the surface crust and the outcrop subsurface and its main sedimentological characteristics. All soil samples were thoroughly wrapped in cling film after sampling to avoid subsequent contamination. We
applied a multi-methodological approach to identify different mineral phases, (micro-)structures, and sedimentological characteristics of wedge and crust samples.

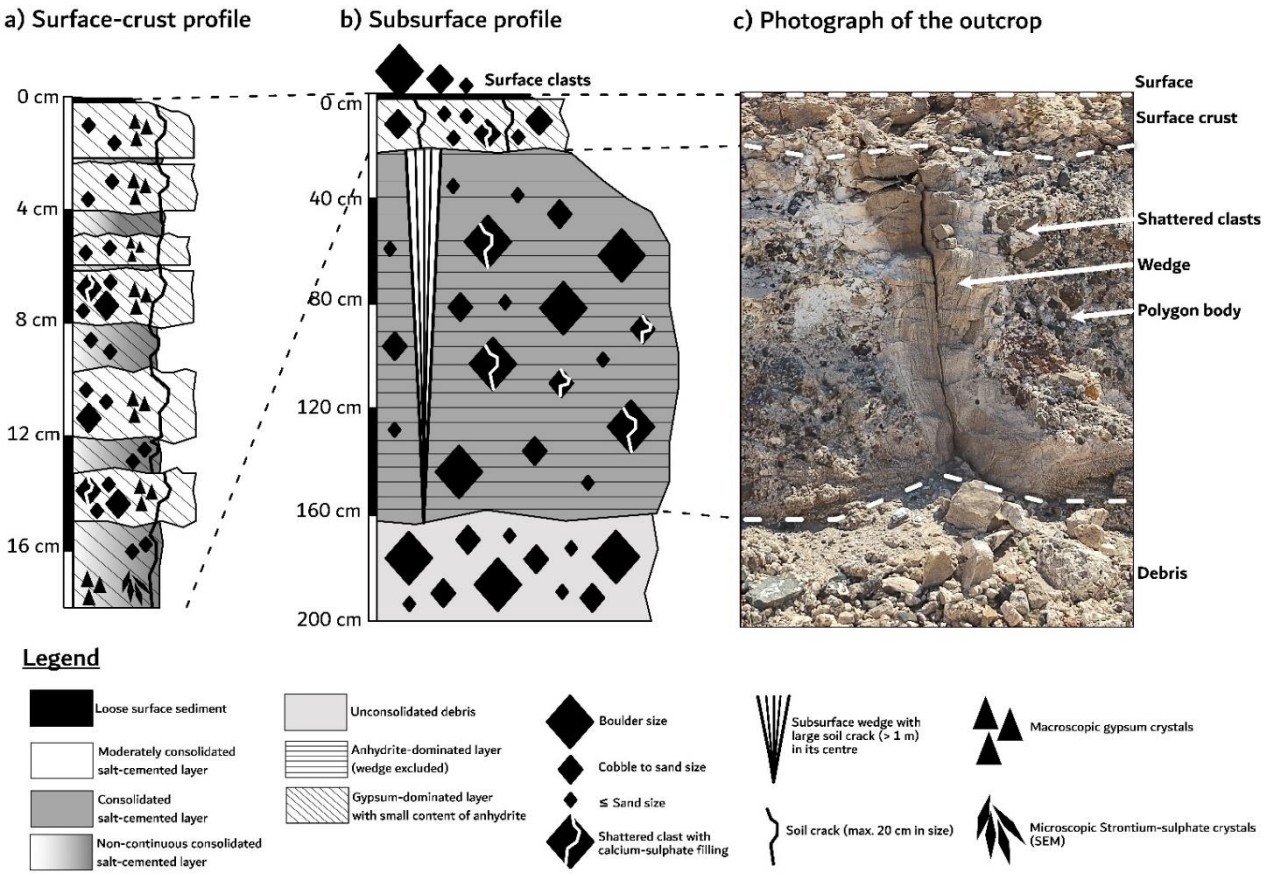

**Figure 2: a) Schematic profile of the ~20 cm thick surface crust at the Aroma site showing its main sedimentological characteristics. The surface crust is characterised by moderately consolidated layers exhibiting macro-crystalline**
**gypsum crystals, which are interrupted by non-continuous consolidated layers ("lens-like") consisting of**

microcrystalline gypsum-dominated cement. The crust shows large horizontal and vertical cracks (up to ~20 cm) partly containing loose cobbles. SEM analysis revealed microscopic lenticular $SrSO_4$ crystals only in the base layer of the crust. b) Schematic subsurface profile of the outcrop at the Aroma site depicting main characteristics; desert pavement, surface calcium sulphate crust, calcium sulphate-cemented matrix with incorporated pebble to boulder sized clasts (representing the 'polygon body') containing a network of vertical cracks and vertically laminated calcium sulphate wedges. Shattered clasts with calcium sulphate fillings occur within the polygon body. c) Photograph of the outcrop.

### 3.1 Geochemical, mineralogical, and sedimentological analyses

All mineralogical, geochemical, and sedimentological analyses of calcium sulphate wedge and crust subsamples were performed at the Institute of Geology and Mineralogy, University of Cologne, Germany. Since the extraction of individual very fine wedge laminae did not provide enough material for the analyses, small wedge parts (sets of laminae) were sampled with a hammer and chisel along the horizontal axis of the wedge ARO18-08 from the wedge centre to the periphery. Figure 4 shows the sample ARO18-08 representing a horizontal cross section, or transect, of the wedge spanning between the periphery and centre of the wedge. The sample was fractured during sampling and hence consists of a left part (bordering the wedge periphery, hereafter abbreviated LP), and a right part (bordering the wedge centre, RP). Subsamples of the crust sample ARO18-02 were taken from the crust surface downwards to the base of the crust at ~18–20 cm (see Fig. 5).

The mineralogical composition of samples was determined by powder X-ray diffraction (XRD) analyses using a Bruker D8 AXS DISCOVER diffractometer with CuKα radiation (λ = 1.54058 Å) operating at room temperature. The patterns were collected between 7 and 120° 2θ with a step size of 0.010° 2θ and a dwell time of 1 s. For quantitative analysis, all samples were refined by a whole powder pattern fitting using the Diffrac.TOPAS (Version 4.2) program with a Pearson VII function for profile fitting. The used database is Power Diffraction File-2 (PDF-2). The XRD subsamples were hand grinded with an agate mortar to minimise phase transitions of calcium sulphate due to temperature increase during sample preparation.

Prior to subsampling wedge sample ARO17-03A was used for high-resolution XRF scanning and radiographic imagery to resolve the fine vertical lamination (see Fig. 4). The wedge ARO17-03A was scanned with an ITRAX core scanner from CoxAnalytical Systems (Croudace et al., 2006) equipped with a Cr-tube using a scan resolution of 200 µm, a voltage of 30 kV and a current of 155 mA with an exposure time of 20 seconds.

Scanning electron microscopy (SEM) was performed on wedge ARO18-08 (2x LP, 1x RP) and crust sample ARO18-02 to analyse microstructures and calcium sulphate cement using a Zeiss Sigma 300-VP equipped with an energy dispersive X-ray spectroscopy (EDX). Prior to the SEM and EDX analysis, the samples were cut, embedded in an epoxy-resin puck of 2.5 cm in diameter, and subsequently gold-sputtered (see sample puck examples in Fig. S.10 in the supplementary material).

Grain-size analysis was conducted using a laser particle analyser from Beckman Coulter LS13320. Prior to sample preparation, the calcium sulphate cement was removed from the bulk sample of wedge ARO18-08 and

surface crust ARO18-02 by dissolving the samples in 10% NaCl solution for 7–10 days. After treatment with the 10% NaCl solution, the clastic material was examined under a microscope to check for any remaining calcium sulphate in the sample or for coatings on the grains. The clastic material of the samples was subsequently treated with 5% $H_2O_2$ to remove any potential organic content and 10% HCl was used to remove carbonate before the sample was dispersed in a 2.5% sodium polyphosphate solution.

Inductively coupled plasma optical emission spectroscopy (ICP-OES) analysis was performed using an ARCOS ICP-OES with an axial plasma observation from SPECTRO Analytical Instruments. Sample water extraction of subsurface wedges ARO17-03A and ARO18-08 as well as surface crust sample ARO18-02 followed the procedure described by Voigt et al. (2020). This sample water extraction aimed in the extraction and quantification of soluble salts in the sample material. The procedure is based on 100 mg sediment, which was leached in deionized water (18.2 MΩ•cm) for 14 days at 25°C to extract the soluble salts from the samples. The concentrations of Ca and S were not considered for analysis as calcium sulphate phases were dissolved to saturation levels in the leachates.

Photogrammetric 3D reconstructions of subsurface wedge (ARO17-03A) and surface crust (ARO18-02) samples were created from image datasets taken in a lightbox environment with a fixed physical camera position and a turntable (see Table S.7 and further information in the supplementary material for more details). Final watertight and scaled meshes (~10 M faces) were used to quantify the specimens' volumes and to determine the bulk density of the samples.

## 3.2 Dating methods

Surface exposure dating was conducted following the noble gas extraction procedure of Ritter et al. (2021) to measure the concentration of cosmogenic $^{21}$Ne in six surface quartz clasts (ARO17-01A–F) from the Aroma fan surface above the studied wedge network (see Fig. 3). During the sampling process in the field, we took care to avoid sampling clast fragments sourced from 'Kernsprung' (insolation weathering) and sampled in an area of approx. 40–50 m$^2$. Samples were crushed, dry sieved by hand to the 250–710 μm fraction, and etched multiple times in HCl and a HF-HNO$_3$ mixture (Kohl and Nishiizumi, 1992). The $^{21}$Ne exposure ages shown in Figure 3 (relative probability plot) are based on the LSD$_n$ scaling scheme of Lifton et al. (2014) and calculated with the CRONUS-Earth online calculators (version 3; https://hess.ess.washington.edu/math/v3/v3_age_in.html) as published by Balco et al. (2008). Cosmogenic nuclide and calculation data as well as $^{21}$Ne triple isotope diagram for the respective exposure ages are provided in the supplementary material (Fig. S.4 and S.5; Data tables: Table S.1 and S.2).

$^{239}$Pu analysis was performed to trace any recent (i.e., Anthropocene) transport of surface sediment into the subsurface through the surface crust. Plutonium subsamples were collected from the calcium sulphate crust

ARO18-02 (see Fig. 5). First, we sampled the dust covering the top surface of the crust using a clean brush (ARO18-02-001 and replicate sample ARO18-02-TC2). Afterwards, we sampled sediment from a cavity located ~10 cm below the surface of the crust (ARO18-02/Pu5) using a long spatula and a vacuum grain picker with a mounted cannula tip. Note that the sample ARO18-02/Pu5 was taken from a different location than indicated in Figure 4 (see Fig. S.9 in the supplementary material for the exact subsample position). The cavity was not exposed at the outcrop, minimising the possibility of pre-sampling contamination due to the road construction works. To avoid sampling of potentially contaminating dust particles (e.g., particles blown in during sample transport), we sampled surfaces located deeply inside the cavity (>4–5 cm behind the cavity opening). Afterwards, we cleared a vertical profile along one side of the crust block by removing ~1–3 cm of the outer surfaces (see Fig. S.8 in the supplementary material) using a handheld rotary tool with a steel blade mounted. Along this profile, we sampled three blocks of 1.5 cm thickness each along the profile (ARO18-02/Pu2: 0–1.5 cm below the horizontal top surface; ARO18-02/Pu3: 1.5-3.0 cm; ARO18-02/Pu4: 3.0–4.5 cm) to investigate downward migration of dust particles inside the heterogenous dense crust. The fragility of the crust block material required a top-down sampling strategy, introducing a certain risk of contamination of the deeper sample with material falling down from above. We mitigated that risk by constantly vacuum-cleaning the surfaces and narrowing the horizontal cutting area at depth. After chemical processing (see supplementary text for details), the samples were measured at CologneAMS (Dewald et al. 2013). Measuring $^{239}$Pu using accelerator mass spectrometry (AMS) bears the advantage of high measurement accuracies achievable for small quantities of sample material (for a comprehensive overview see e.g. Alewell et al., 2017). We further attempted to quantify $^{240}$Pu from the same sampling material, but an unusual piling of counts at the targeted mass per charge ratio caused most measurements to be unreliable (exception: ARO18-02-TC2).

## 4 Results

### 4.1 Calcium sulphate wedge analyses

The investigated outcrop extends to a depth of ~1.8–2 m. Below the surface crust are clasts ranging in size from pebbles to boulders in a fine-grained calcium sulphate matrix ('polygon body'; see Fig. 2). Many clasts are shattered, and cracks are mainly filled with calcium sulphate and 30.4 wt% aluminite in clast ARO18-04 and 17.6 wt% aluminite in clast ARO18-05 (see Fig. S.3 in the supplementary material). The polygon bodies comprise a network of large soil cracks (vertical extent >1 m, see Fig. 2b, c) and adjacent, vertically laminated parts left and right of the crack, which represent the calcium sulphate wedges. The base of the trench outcrop is covered by debris. $^{21}$Ne surface exposure ages of the surface clasts vary from 3.3 ± 0.3 to 5.4 ± 0.4 Ma with a mean age of 4.34 ± 0.36 Ma (see Fig. 3, further detailed information in the supplementary material).

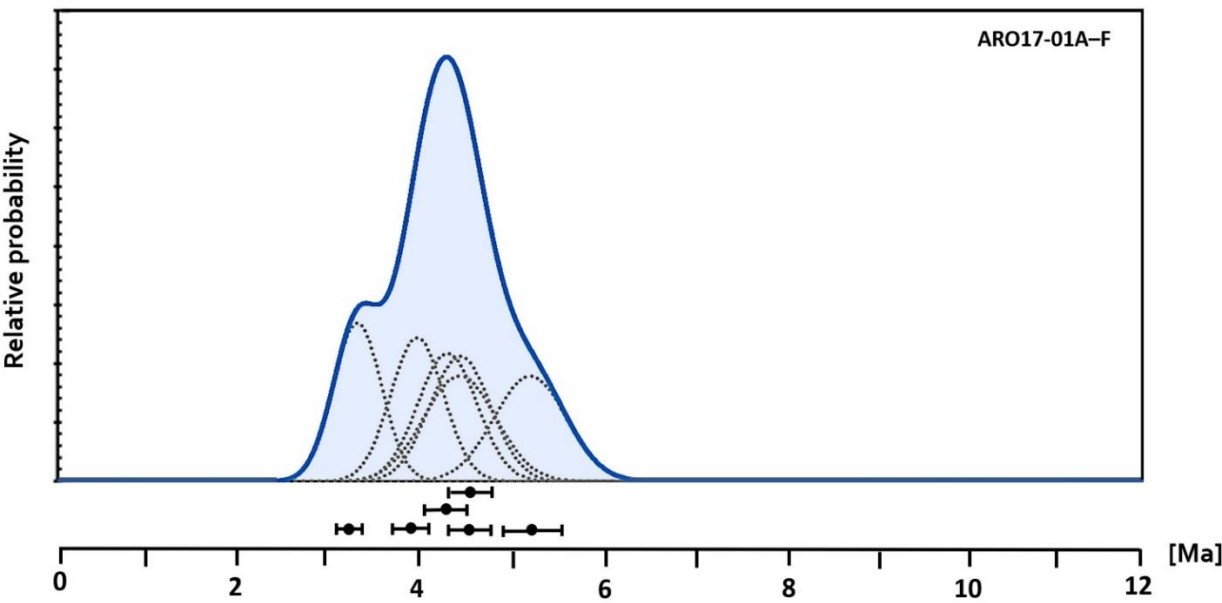

295**Figure 3: Relative probability functions of individual surface quartz clasts (ARO17-01A–F; grey dotted) and cumulative curve (blue shaded). Error bars (±1σ) are based on the external errors (see Table S.2 in the supplementary material). The cumulative curve indicates a distinct peak at ~4.3 Ma.**

XRD results of salt precipitates from two shattered clast samples (ARO18-04 and ARO18-05) revealed ~70 wt% of β-anhydrite, ~20–30 wt% of other evaporites (mainly aluminite), and ~0.5–15 wt% clastic material in the 300    cement matrix (see Fig. S.3 in the supplementary material). The group of other evaporites comprises up to 17–30 wt% aluminite ($Al_2SO_4(OH)_4 \cdot 7H_2O$), and traces of konyaite ($Na_2Mg(SO_4)_2 \cdot 5H_2O$) and halite (NaCl) (see Table S.4 in the supplementary material).

The vertical lamination of the calcium sulphate wedges (Fig. 4; laminated part) extends from the centre of the wedge (soil crack in the outcrop) to the periphery (polygon body direction). Lamination of the wedges is less 305    distinct at the periphery (see Fig. 4B and 4C; radiographic image). Wedge sample ARO17-03A was used to calculate the bulk density from the photogrammetrically derived 3D model, yielding a density of 1.68 ± 0.04 g/cm$^3$ (see supplementary material for more details).

The (evaporite-free) clastic sediments in wedge ARO18-08 are dominated by sand, with medium and fine sand being the most abundant grain sizes (Fig. 4 B.3). SEM images taken at three different positions along the wedge 310    transect reveal different densities of calcium sulphate cementation, with the general pattern of increasing densities towards the periphery and decreasing densities towards the centre of the wedge. The SEM images taken from the centre of the wedge and close to the periphery show that the cementation density varies randomly within the wedge depending on the cement content of the individual fine laminae. The generally high content of

calcium sulphate cement varies between ~40–70 wt% and the clastic content varies between ~26–50 wt% at the
periphery and ~50–60 wt% at the centre of the wedge (see Fig. 4 B.2; XRD results). The Ca and S compositions
in wedge ARO17-03A (Fig. 4C) match, and increase towards the periphery, in particular within the non-laminated
part of the wedge. The XRD results of wedge ARO18-08 indicate that gypsum is mainly distributed in the part
near the wedge centre, reaching the highest gypsum content in subsample 1 and 2 at the wedge centre with ~40
wt% gypsum. The dominant calcium sulphate phase of the wedge is anhydrite with up to ~73 wt% in the part near
the periphery, decreasing to a minimum abundance (~3 wt%) in the subsamples close to the wedge centre (see
Table S.4 in the supplementary material). The Na and Cl concentrations based on ICP-OES results show that Na
and Cl were both dissolved in the leachates during the sample water extraction procedure (see Table S.3 in the
supplementary material). The increased Na concentration indicates that halite but also other sodium- and
chloride-bearing soluble salts could be present in the samples, but are not resolvable in our ICP-OES data.
However, the XRD results of some wedge subsamples show traces of other sulphates occurring besides calcium
sulphate such as aluminite, arcanite ($K_2SO_4$), amarantite ($FeSO_4(OH)\cdot 3H2O$), and peretaite
($Ca(SbO)_4(SO_4)_2(OH)_2$) (see Table S.4 in the supplementary material).

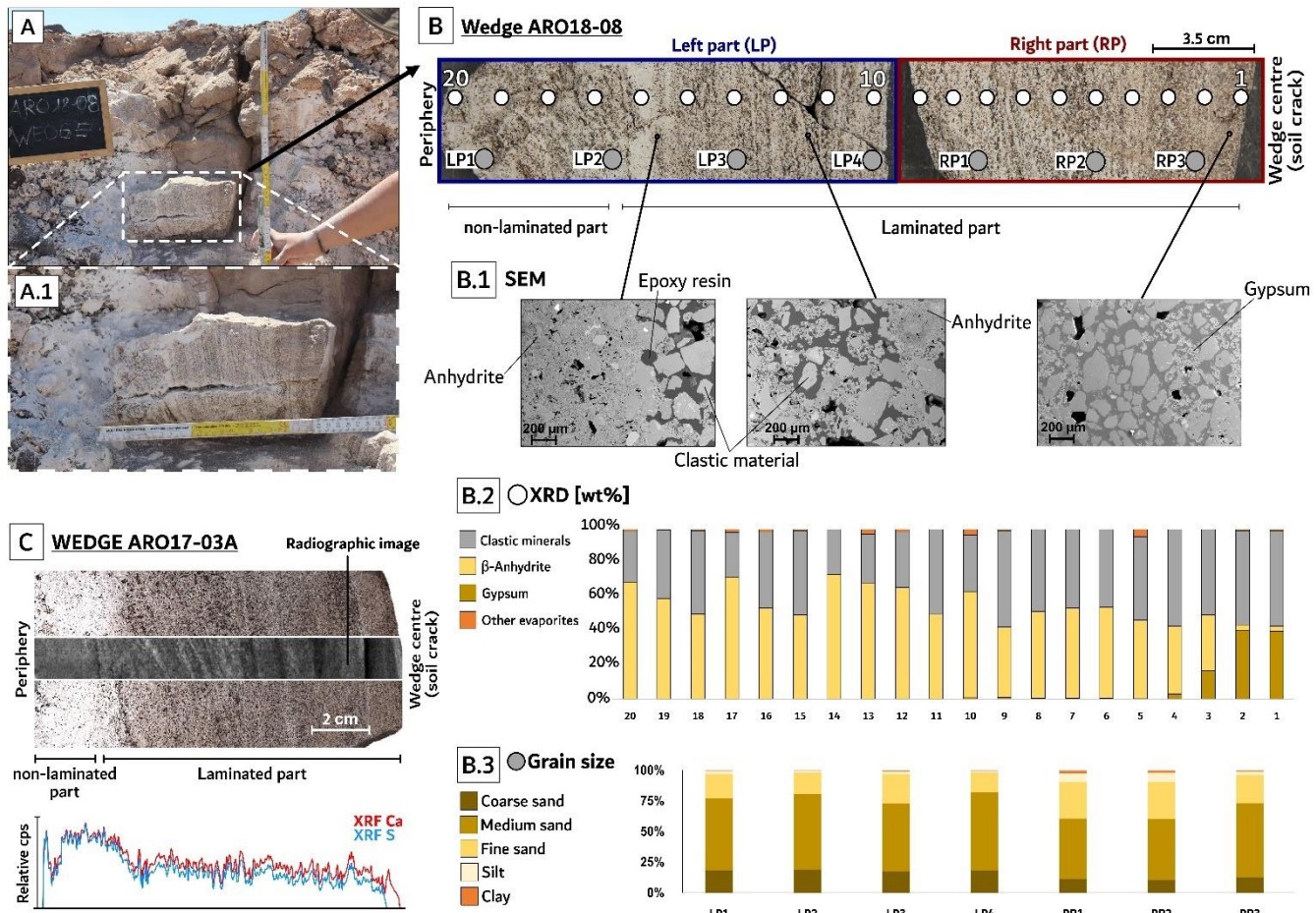

**Figure 4: A) Outcrop image and close-up (A.1) of wedge ARO18-08 sampled from the outcrop wall at ~40–50 cm depth. B) Photograph of vertically laminated wedge ARO18-08 showing laminated and non-laminated sections and the outcrop orientation of the wedge (right: wedge centre; left: periphery). The positions of the XRD and ICP-OES subsamples (see Tab. S.3 in the supplementary material for ICP-OES results) are indicated by white circles and the subsamples for grain size analysis are indicated by grey circles. B.1) SEM images of three positions within the wedge sample showing different densities of calcium sulphate cementation. B.2) XRD results of 20 wedge subsamples (white circles). B.3) Grain size results of wedge subsamples (grey circles). C) Photograph and radiographic image of wedge ARO17-03A. The XRF results show a match between Ca and S throughout the wedge. The very fine lamination is also visible in the fluctuations of the XRF Ca and S contents.**

## 4.2 Surface crust analyses

The surface crust represents the top ~20 cm of the studied outcrop (see Fig. 2 and Fig. S.7 in the supplementary material). The crust surface is covered by dust and a large quantity of clastic material larger than sand, typical for an unconsolidated desert pavement (see Fig. 2 and Fig. S.2 in the supplementary material). Below the surface, the crust is moderately cemented, predominantly by gypsum. The bulk density derived from the photogrammetric

3D model of the sample block ARO18-02 is 1.34 ± 0.03 g/cm$^3$, based on the surface crust mass of 4031 ± 0.5 g

and a volume of 3014.29 ± 70.24 cm$^3$ (see supplementary datasets for more details).

Grain size data from clastic material of four crust subsamples indicate a dominance of the medium to fine sand fraction across the crust (Fig. S.12). Microscopic images show fibrous macrocrystalline gypsum crystals on consolidated wavy and partly 'nodule-like' microcrystalline calcium sulphate cement (Fig. 5; Fig. S.12). The crust surface contains ~43 wt% clastic minerals, ~35 wt% of aluminite ($Al_2SO_4(OH)_4 \cdot 7H_2O$), and traces of gypsum

(~0.8 wt%) as revealed by the XRD measurements (see Table S.4). Subsamples taken from 3–17 cm below the surface (ARO18-02-002 and all samples below) show a considerable change in mineralogy as the gypsum content increases up to ~70–90 wt % while the clastic mineral content decreases to <11 wt% with increasing depth. The β-anhydrite content increases with increasing depth from ~1 wt% in the upper part to up to ~24 wt% in the bottom crust sample ARO18-02-007. The Na and Cl concentrations based on ICP-OES results are below the

detection limit for all crust subsamples except for the bottom sample ARO18-02-007 (0.77 mol/l Na), indicating a general absence of NaCl in the surface crust. The presence of other sulphates is confirmed by XRD results, which indicate the presence of aluminite, alunogen ($Al_2(SO_4)_3 \cdot 17H_2O$), konyaite, and ramsbeckite (($Cu,Zn)_{15}(SO_4)_4(OH)_{22} \cdot 6H_2O$) (see Table S.4). Subsample ARO18-02-007 shows microcrystalline lenticular crystals of celestine ($SrSO_4$) as illustrated by the SEM and EDX element distribution images (Fig. 5; SEM3).

The crust is characterised by a generally high porosity and large cracks (>15 cm in depth) containing gypsum crystals and pebble to cobble-sized clasts (see crust photo in Fig. 5). The highest blank-corrected $^{239}$Pu concentrations were measured on the crust surface (Fig. 5; ARO18-02-001, 6.48 ± 0.20 mBq kg$^{-1}$, ARO18-02-TC2, 6.09 ± 0.23 mBq kg$^{-1}$) and in the sampled cavity (ARO18-02/Pu5; 8.68 ± 0.59 mBq kg$^{-1}$). These values are well in range of what has been measured close to the city of Iquique at similar latitudes (~20°S, ~7

mBq kg$^{-1}$ assuming a soil density of 1.8 g cm$^{-3}$ and an isotope ratio of ~0.17; Chamizo et al., 2011). Due to low count rates, blank correction amounted to >20% for $^{239}$Pu measurements of subsamples ARO18-02/Pu2, ARO18-02/Pu3 and ARO18-02/Pu4 (see Table S.5 in the supplementary material). We consider these blank subtraction values as being too high to draw any detailed conclusions from the individual blank-corrected concentrations. However, the low count rates reflect extremely low nuclide concentrations (blank-corrected

concentrations <1.19 ± 0.15 mBq kg$^{-1}$) in these subsamples. We report a similar blank subtraction of 22% for the $^{240}$Pu measurement of ARO18-02-TC2, which is related to high blank levels. While similar constrains on the blank-corrected $^{240}$Pu concentration of this sample apply as valid for the other subsamples with high relative measurement background levels, it is worth to note that the resulting $^{240}$Pu/$^{239}$Pu is 0.185 ± 0.020. This ratio reflects the global fallout signature, i.e. the source of the plutonium measured in the samples is likely to be

originating from the atmospheric weapon tests conducted during the 1950s and 1960s (i.e. $^{240}$Pu/$^{239}$Pu = 0.173 ± 0.027 for 0-30°S; Kelley et al., 1999; 0.166 ± 0.008 for Iquique at ~20°S, Chamizo et al., 2011).

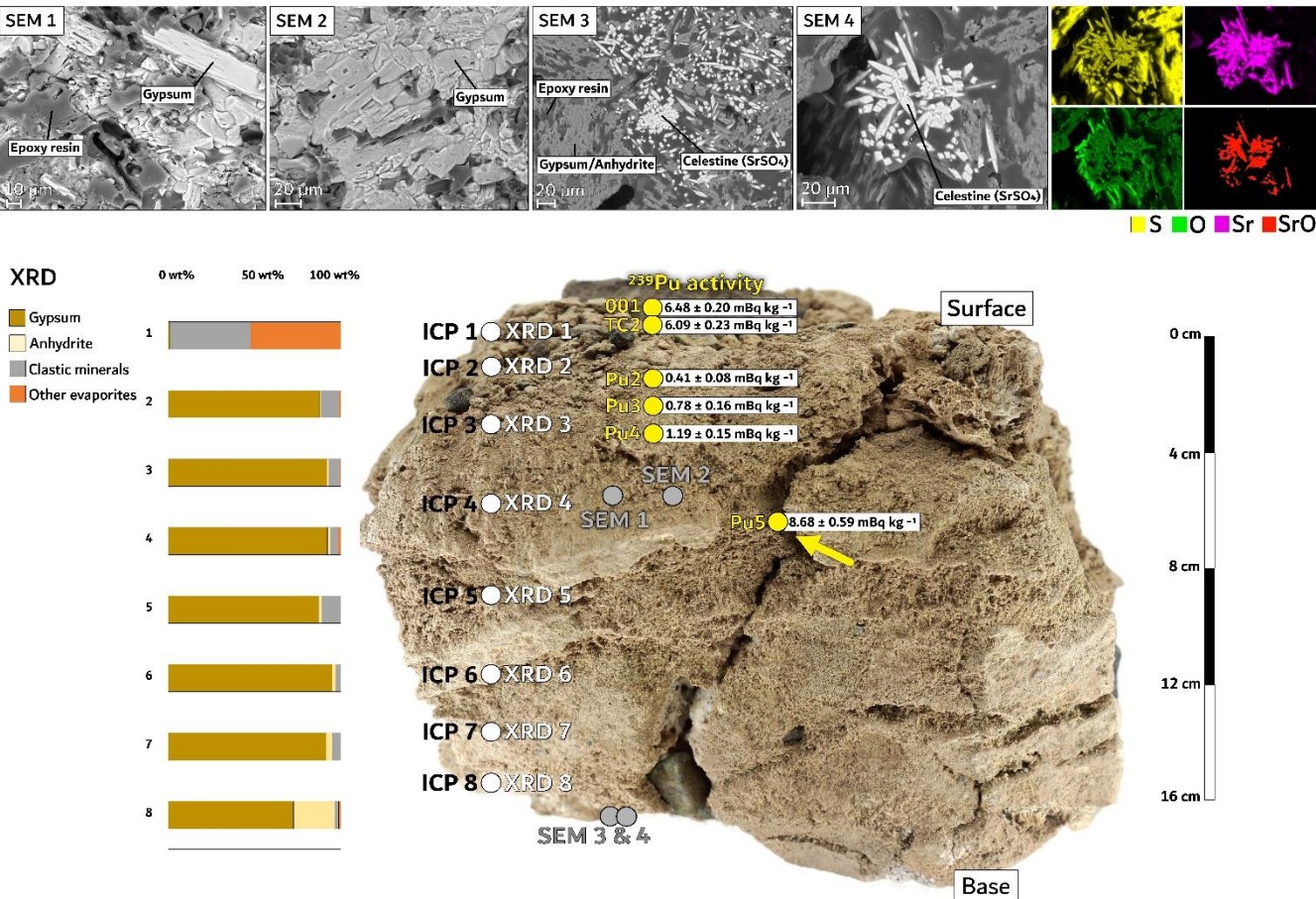

**Figure 5: Photograph of ARO18-02 (surface crust) and compilation of applied analyses. The surface crust is**
**characterised by a high clastic mineral and aluminite content (except for calcium sulphate) on the surface and a**
**generally high gypsum content in the subsamples below the surface (see XRD results; white circles). The cement**
**includes microcrystalline gypsum crystals (see SEM1 and 2; grey circles). SEM3 and SEM4 show lenticular crystals**
**of celestine (SrSO₄), which occurs only in the crust base subsample ARO18-02-007. The coloured SEM images**
**display the distribution of elements (S, O, Sr) and the oxide SrO, highlighting that these structures are indicative of**
**celestine. The ²³⁹Pu activities are presented by yellow circles. The highest ²³⁹Pu activities are in the surface and**
**crack interior subsamples.**

## 5 Discussion

### 5.1 Formation hypothesis of subsurface wedges and polygonal patterned ground

The high calcium sulphate content of the investigated subsurface wedges contrasts with previous descriptions of
wedge-polygon-structures in the Atacama Desert. This finding implies that calcium sulphate-driven salt dynamics
may be a dominant factor in the formation of wedges and polygons at the Aroma fan site, rather than thermal

contraction (low-salt sand wedges; Sager et al., 2021) or polygon formation in playa-like environments due to desiccation (e.g. Ericksen 1981,1983; Bobst et al., 2001; Finstad et al., 2016).

The Aroma fan wedge-polygon formation has developed in and on alluvial fan deposits, which contrasts with the
polygon formation observed in a playa environment (e.g. Cheng et al., 2021; Zhu et al., 2024). While the Aroma fan wedges share a similar desiccation mechanism (calcium sulphate dehydration) with desiccation polygons in playa environments, the surrounding environment and deposits are distinct. Although thermal contraction processes cannot be entirely ruled out as a contributing factor in the formation of the wedge structures at the Aroma site, this mechanism is likely to play a minor role. The high calcium sulphate content in the Aroma fan
wedges indicates that desiccation due to dehydration of gypsum is the most likely process responsible for the local wedge formation.

Previous studies suggested that dissolution and precipitation of salts are the most important processes contributing to salt heave processes in the Atacama Desert (Buck et al., 2006; Howell, 2009) and clast shattering (Winkler and Singer 1972; Amit et al., 1993; Rodriguez-Navarro and Doehne, 1999), as the solution
supersaturation ratio is proportional to the crystallisation pressure (Winkler and Singer, 1972), a pattern that is controlled by high evaporation rates and solute availability (Howell, 2009). The dissolution and precipitation processes of calcium sulphate are evident from the high content of the naturally occurring β-anhydrite in the wedge and in the shattered clasts from the Aroma fan outcrop (Fig. 1). It is hypothesised that β-anhydrite is formed exclusively by the precipitation from a highly saline solution at temperatures as low as 60°C (Hardie,
1967; Cody and Hull, 1980). However, the question of whether this formation occurs under ambient desert conditions remains a topic of debate (Ritterbach and Becker, 2020; Wehmann et al., 2023). It has not yet been determined whether the gypsum content in the wedge samples is primary or secondary. The latter is formed by the rehydration of γ-anhydrite over bassanite back to gypsum (Mossop and Shearman, 1973), which would imply that swelling and shrinking processes contribute to haloturbation mechanisms in the subsurface. Shi et al. (2022)
proposed that tunnels in the hexagonal crystal structure of γ-anhydrite from the Atacama Desert can incorporate cations of Si and P, which are thought to attenuate phase transition from γ-anhydrite to bassanite. The authors discussed that this phenomenon enables γ-anhydrite to be prevalent in hyperarid environments such as the Atacama Desert and Mars. Ritterbach and Becker (2020) posited that the dehydration of gypsum to bassanite and subsequently to β-anhydrite may require long periods of time at temperatures of 80 °C and even lower, which
may explain the presence of β-anhydrite in deposits from hyperarid environments.

The salt cementation of subsurface wedges in the Atacama Desert has also been discussed in previous studies. Sager et al. (2021) observed that the outer parts of sand wedges from the Yungay region exhibited higher salt concentrations relative to the inner parts. They suggested that, over time and with repeated rainfall, salts migrate from the salt-rich polygons to the initially salt-poor wedges, eventually causing their cementation. Furthermore,
Sager et al. (2023) conducted a rain experiment, which demonstrated that salt precipitation occurs on the surface

of the wedges after wetting. This is likely due to the upward movement of saline water along the wedges. Although we cannot completely rule out the possibility of post-formation cementation of the wedges, the cementation of the Aroma fan wedges may have resulted from calcium sulphate infiltration into the host sediments prior to the formation of the wedges, or alternatively directly from the surface soil. However, the available data do not allow a distinction to be made between these sources. As noted above, beta-anhydrite is both insoluble and stable under the current environmental conditions of the Atacama Desert. While some movement may occur from areas of high to low concentration, the mobility of calcium sulphate is significantly reduced once stable beta-anhydrite is formed.

Pfeiffer et al. (2021) conducted water infiltration in calcium sulphate-rich soils across different sites in the Atacama Desert. The study revealed a consistent sequence of soil horizons at all sites, characterised by a highly porous and conductive anhydrite layer above an impermeable, cemented gypsum layer. Significant water infiltration occurs mainly through the porous, conductive layer (such as the "chusca" layer at the Yungay site), while in the gypsum-cemented layer, infiltration is limited to vertical polygonal cracks, which are approx. 1.5 meters deep (as observed in the petrogypsic layer at Yungay). The processes of infiltration, dissolution, and reprecipitation of calcium sulphate at the Aroma fan site are thought to be concentrated around wedge structures, particularly within the cracks. The recent movement of fine particles, mainly through the cracks in the calcium sulphate-rich surface crust, as indicated by Pu isotopes, can be considered as a modern analogue of these processes.

The hyperarid soil genesis model proposed by Howell (2009) can be applied to understand wedge formation mechanisms in our study area (Fig. 6). Based on this model, a sequence of wedge formation could begin with the delivery of meteoric water (infrequent rain event), followed by infiltration into the coarse-grained and poorly sorted alluvium of the Aroma fan site. During infiltration, it is assumed that meteoric water dissolves soluble salts at the surface and in the soil, and thus supports the downward infiltration of these salts (see Fig. 6, step 1). At greater depths, the saline solution exceeds saturation and salts precipitate in the pore space of the alluvium (Fig. 6, step 2). The resulting destructive crystallisation pressure of the precipitated gypsum or anhydrite causes significant mechanic damage in the surrounding deposits as reported in previous studies (e.g. Buck et al., 2006; Howell, 2009; Benavente et al., 2006; Schiro, et al., 2012; Flatt et al., 2014). Fracturing processes in the Aroma fan outcrop are reflected by numerous soil cracks in the polygon as well as fractures in cobble to boulder sized clast (see Fig. 1 and S.1, S.3 in the supplementary material). Stress caused by subsurface volume increase in the polygon body leads to preferential deformation along the axis of least resistance, i.e. upwards, and is referred to as salt heave processes (Buck et al. 2006).

The surface sediment is deposited in soil cracks formed by desiccation processes (predominantly dehydration of gypsum). A specific sediment transport mechanism cannot be determined as the well-sorted grain size distribution of the wedge material (mainly medium to fine sand fraction, Fig. 4) is indicative of aeolian deposition,

but due to the potential for intermittent rainfall events at the Aroma fan site, low-magnitude fluvial transport (confined to the debris on top of the wedge) cannot be excluded.

Repeated cycles of frequent moisture events, or intermittent phases thereof, may have caused the accumulation of calcium sulphate within the soil crack. Swelling (hydration) and shrinking (dehydration) processes due to the phase transformation of gypsum to $\gamma$-CaSO$_4$ and vice versa could have led to an increase in crack width and depth, as well as increased clast fracturing (see Fig. 6, step 3; cf. Howell, 2009). As a result, surface sediments, salts and moisture can rapidly infiltrate to greater depths, and enlarged soil cracks act as 'salt and moisture conduits', allowing haloturbation to occur in even deeper deposits (Howell, 2009).

Salt heave processes intensify as moisture events and haloturbation processes are repeated over time, gradually forming a microtopographic signature that represents a polygonal patterned ground at the surface (see Fig. 6, step 3; cf. Buck et al. 2006; Howell, 2009). Note that the purple dashed lines in step 6 (Fig. 6) only show a hypothetical polygonal patterned ground beneath the surface crust, as there is no evidence of a microtopographic signature at the base of the surface crust sample at our study site. Repeated haloturbation (swelling and shrinking) leads to tensile stresses in the cohesive material and develops expansion and contraction forces, resulting in reopening of cracks that are refilled during the next depositional cycle (Howell, 2009). This mechanism has probably caused the formation of a vertical lamination of the Aroma fan wedges, consisting of calcium sulphate and clastic-dominated sediment (cf. Howell, 2009). Apart from visual interpretation, the lamination is evident from the XRD and XRF results of both wedges, as well as the radiographic image of wedge ARO17-03A (Fig. 4, 6.4, 6.5). As shown in Figure 4B and 4C, both analysed wedges show a non-laminated and $\beta$-anhydrite-dominated part at the wedge periphery, the origin of which is not yet clear. This part could either represent the stratigraphically oldest part (early precipitation of calcium sulphate without surface sediment input) or it could indicate that the layers close to the periphery and the polygon body material were homogenised during intensified haloturbation (repeated dissolution and reprecipitation, and/or swelling and shrinking processes) stresses in the subsurface. Repeated haloturbation processes have enhanced the cementation of these parts, supported by additional saline solutions dissolved from the calcium sulphate-rich surface soil. The dominance of $\beta$-anhydrite is presumably caused either by direct precipitation from highly saline solutions or by dehydration of gypsum to bassanite on to $\beta$-anhydrite, provided that the phase transition occurs over a long period of time (Ritterbach and Becker, 2020). The increased gypsum content in the subsamples ARO18-08-RP1–4 directly at the wedge centre (soil crack side) indicates a phase transition from gypsum to anhydrite, likely to correspond to the increasing age of the wedge. This process is associated with a volume decrease of ~29% for the gypsum-bassanite transition (Milsch et al., 2011) and a total of ~39% for the gypsum-anhydrite transition (Milsch et al. 2011; Sanzeni et al. 2016). Such volumetric changes could result in a shrinkage of laminated deposited sediment in previously open cracks, leading to reopening of the soil crack. Thus, we assume that shrinking processes dominate in the wedge centre, causing repeated reopening of the soil crack.

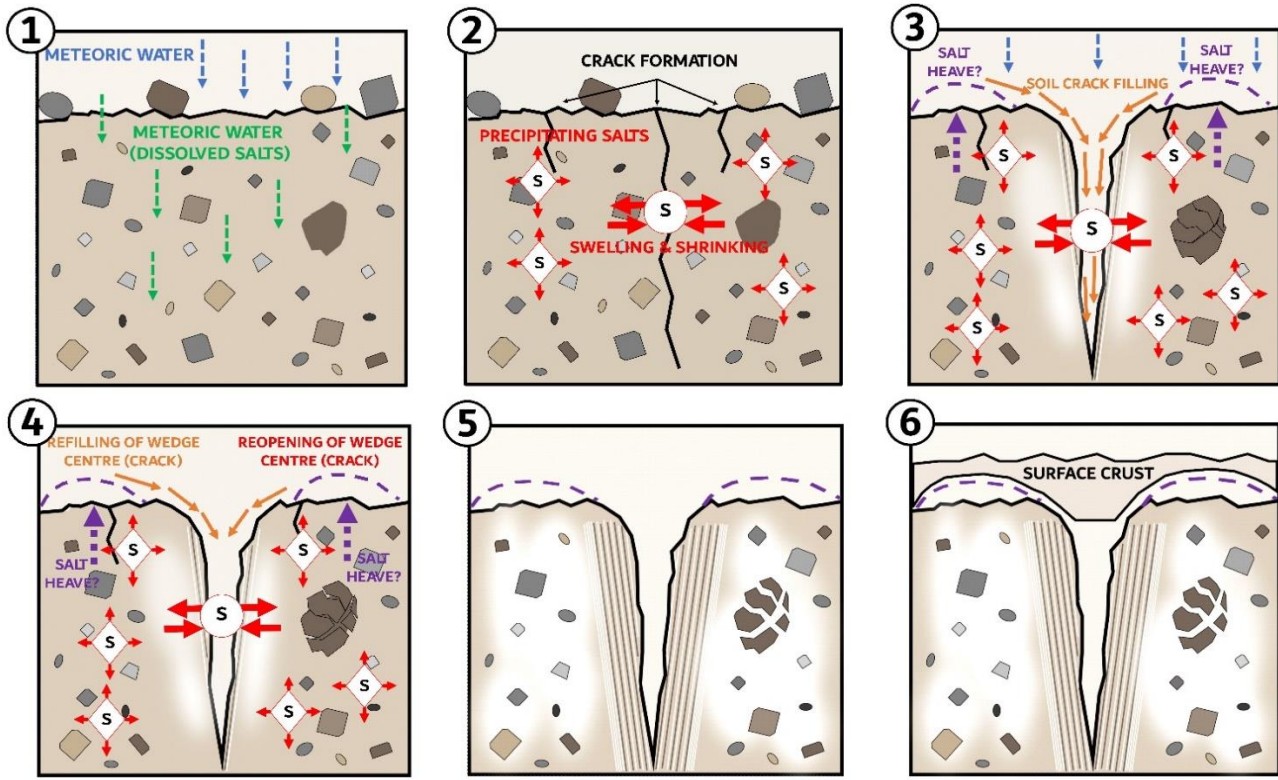


**Figure 6: Sequence of wedge formation processes in the subsurface of the Aroma fan site based on the hyperarid soil genesis model of Howell (2009). 1) Meteoric water infiltrates porous alluvium and dissolves soluble salts, which precipitate in the pore space and indurate the alluvial sediment. 2) Haloturbation (e.g. swelling/shrinking as white circle and dissolution/precipitation of calcium sulphate as white rhombuses) creates destructive pressure in the**
**subsurface, leading to the formation of soil cracks and subsequent filling of the soil cracks with (surface) sediment. Significant subsurface pressure results in salt heave (Buck et al. 2006), where the subsurface pressure is released upwards, as this is the direction of least resistance (purple dashed arrows). 3) Meteoric water re-infiltrates the alluvium and the processes of 1) and 2) are repeated, resulting in soil crack growth (both in width and depth) and clast shattering in the sediment. Haloturbation processes and subsequent reopening of soil cracks promotes the**
**infiltration of sediment, salts and moisture to greater depths. 4) Multiple cycles of haloturbation (dominated by swelling and shrinking due to dehydration of gypsum or hydration of γ-anhydrite) cause reopening and refilling of cracks. 5) The product of long-term haloturbation processes in the subsurface associated with multiple moisture events is the characteristic vertical lamination of calcium sulphate wedges as well as a polygonal patterned ground (microtopographic signature, purple dashed lines) on the surface due to salt heave mechanisms. 6) Probably after**
**wedge-polygon-formation, the surface crust covers the polygonal patterned ground, indicating an environmental change that favoured the formation of the crust. The purple dashed lines represent only a hypothetical polygonal patterned ground, as the surface crust sample shows no evidence of a microtopographic signature at its base.**

## 5.2 Formation hypothesis of the surface crust

Calcium sulphate-rich soils and surface crusts in the Atacama Desert have been described in numerous previous
studies (e.g. Rech et al., 2003; Ewing et al., 2006; Wiezchos et al. 2010; Wang et al., 2015; Rech et al., 2019; Ritter et al., 2022). These calcium sulphate-rich soils and crusts are formed by atmospheric deposition (Rech et

al., 2003; Wang et al., 2015; Rech et al., 2019) and contribute significantly to landscape protection against erosion (e. g. Mohren et al., 2020; Ritter et al., 2022), and cover large areas of the Atacama Desert (e.g. Hartley and May, 1998; Ericksen 1981,1983; Rech et al., 2003; Ewing et al., 2006; Rech et al., 2006; Ritter et al., 2022).

The $^{21}$Ne exposure ages (4.34 ± 0.36 Ma) of the desert pavement quartz clasts from the Aroma fan surface could imply that the pebbles and cobbles, which remained stationary at the surface after deposition, were lifted by blown-in accretionary dust ('born at the surface model'; Wells et al., 1995). Our $^{21}$Ne surface exposure ages broadly match the surface formation pulses at ~7 Ma and ~3 Ma, as described by Evenstar et al. (2009). It should be noted that these authors measured $^{3}$He from ~0.5–1 m boulders (including samples taken in the vicinity of our

sampling site) and used a different scaling scheme (the St scaling scheme; Lal 1991 and Stone 2000, for a comparison of scaling schemes for our dataset see supplementary material Table S.2).

These age ranges indicate the end of major alluvial deposition and the onset of significant accumulation of gypsum by atmospheric deposition at the transition from the Late Pliocene to the Early Pleistocene. Assuming that surface exposure ages represent the onset of calcium sulphate-rich soil accumulation (the letter is essential

for wedge formation), we suggest that the formation of wedges, the subsequent cessation of their activity, and the development of a calcium sulphate-rich surface crust capping the wedge structures likely occurred after the Late Pliocene. As the subsurface wedge system developed through multiple moisture events (haloturbation processes), climatic conditions changed from arid to likely hyperarid, with infrequent moisture periods sufficient to sustain wedge activity. Ultimately, a shift to the recent extreme hyperaridity during the Pleistocene caused the

cessation of wedge activity and the net accumulation of atmospheric dust during more drier conditions.

Considering the absence of polygonal patterned ground on the surface of the Aroma fan, we suggest that the surface crust covered the patterned ground and might have attenuated haloturbation processes in the subsurface, as arid to hyperarid conditions favoured the accumulation of gypsum rather than redistribution and secondary modification of gypsum deposits from the surface into the subsurface.

On modern timescales, our $^{239}$Pu data indicate that sediment fines have migrated along cracks towards the inner crust during the past ~70 years. We measure comparably high and consistent $^{239}$Pu concentrations in the surface samples, while the dense crustal parts immediately below the crust surface have $^{239}$Pu concentrations at the detection limit. After deposition, plutonium isotopes adsorb to soil fines, and downward migration of $^{239}$Pu may be controlled by physical processes (for an overview on the environmental behaviour of Pu isotopes see e.g. Alewell

et al., 2017). Consequently, low $^{239}$Pu concentrations within the dense parts of the crust might be expected to contrast with higher $^{239}$Pu concentrations measured within the crust cavity. The apparent relocation of Pu-marked fines to ~10 cm depth during the past ~70 years implies that sediment (and probably moisture) transport to the subsurface is still an active process in recent times. Such a relocation of fines is likely to occur along crust cracks. It remains unresolved whether fractions of surface sediments and/or moisture can pass through the crust

in its present state in the long-term to feed processes forming the wedge-polygon-system.

However, it appears that the wedge-polygon-system, subsurface haloturbation and salt heave forces have become significantly weakened since the formation of the surface crust. The absence of polygonal patterned ground on the Aroma fan surface may be attributed to the inhibition of salt heave processes, as haloturbation is still present but not as intense as prior to crust formation. The attenuated haloturbation forces in the subsurface may have resulted in stress within the surface crust, which subsequently developed numerous cracks thereby facilitating the migration of surface material into the subsurface.

## 5.3 Implications of the formation of other evaporites in the Aroma fan deposits

XRD measurements on samples taken from wedge ARO18-08 revealed traces of hydrated sulphate phases such as aluminite (and potentially other Na- and Cl-bearing salts that could not be distinguished by ICP-OES results from ARO18-08) (see Table S.3 and S.4). This finding may imply that multiple cycles of dissolution and precipitation have caused the dissolution of salt and the alteration of weathering-sensitive minerals such as feldspar, which is abundant in the sample material. Chukanov et al. (2013) first described the mineral vendidaite $(Al_2(SO_4)(OH)_3Cl\cdot6H_2O)$ from a copper mine in the Antofagasta region, which is chemically similar to the hydrated aluminite present in the Aroma fan material. The occurrence of these 'exotic' aluminium-bearing salts is interpreted by Chukanov et al. (2013) as an indicator of feldspar alteration, which is favoured by exposure to sulphuric acid resulting from the oxidation of primary sulphates. Furthermore, Joeckel et al. (2011) concluded that the presence of aluminite (among other Al-bearing sulphates) in the deposits may reflect long-term exposure of rocks or sediments to weathering in natural environments. Aluminite is also detected in the fracture fillings of the shattered clasts in the polygonal body (at a depth of ~110 cm depth below the surface) and makes up ~35 wt% of the surface sediment (surface subsample ARO18-02-001, see diffractogram in Fig. S.6 in the supplementary material). This finding indicates that feldspar alteration occurs at varying depths (surface and subsurface) along the outcrop. In contrast to the copper-dominated study site of Chukanov et al. (2013), and considering the mineralogical composition of the Aroma fan samples, it can be posited that acidic weathering is an unlikely cause of primary sulphate oxidation and feldspar alteration. The high levels of aluminite at the Aroma fan site suggest that feldspar weathering is likely to be induced by sufficient meteoric water over time, mobilising Al from the feldspars in the presence of calcium sulphate.

In contrast to the subsurface wedges, the surface crust mineralogy is dominated by gypsum, with low contents of clastic minerals, anhydrite, and other evaporites, including aluminite, konyaite, and celestine. The presence of sulphates indicates minor dissolution and reprecipitation of salts, the low content of β-anhydrite and the other sulphates suggest that these processes are still active, however, strongly reduced compared to the subsurface wedge system. However, mineral phases such as celestine were only identified as microscopic crystals in the crust's base sample, accompanied by an increased content of β-anhydrite (~23 wt%). This pattern may be indicative of either a dehydration process of gypsum or crystallisation processes following extreme evaporation of

highly saline brines (Waele et al. 2017). Given that β-anhydrite is thought to require highly saline solutions to

precipitate, this is a plausible hypothesis. On the contrary, the absence of halite in the surface crust of the Aroma fan could imply that the highly water-soluble halite was washed out of the crust and migrated downwards to deeper levels during infrequent rain events. Furthermore, Arens et al. (2021) described this phenomenon and proposed that soluble salts could be partially leached out of such hyperarid soils. Given the uncertainty regarding the intensity and frequency of these rain events at the Aroma fan site, it is not yet possible to predict the depth at

which halite may occur in the outcrop. The ICP-OES results of the analysed wedge probably show traces of NaCl, but we suspect that the absence of halite or chlorides in the surface crust, which should be generally present in the study area (e.g. Voigt et al., 2020), is due to rain-induced leaching of highly soluble salts to greater depths in the outcrop (>2 m). Considering that the outcrop was exposed to atmospheric processes for an extended period, possibly lasting weeks to months, a sampling bias cannot be entirely ruled out, although no

signs of post-exposure alterations were observed.

## 5.4 Implications of palaeoclimate and environmental conditions during surface crust formation and wedge growth

The [21]Ne surface exposure ages measured in this study may suggest the end of alluvial deposition and the onset of calcium sulphate accumulation during the Pliocene. Likewise, subsurface haloturbation processes and wedge

growth, as illustrated in Figure 6, may have commenced at that time. It is likely that during the Pleistocene, due to the formation of the surface crust and the associated climatic shift towards drier conditions with fewer wet periods, wedge growth may have attenuated even before a significant crust layer had fully developed. A shift towards hyperarid conditions is believed to have occurred at the transition to the Holocene at the Andean foreslope (e.g. Jordan et al., 2014). Zinelabedin et al. (2022) presented a first approach to applying feldspar

luminescence dating to a calcium sulphate wedge from the Aroma fan outcrop. The widespread equivalent dose distribution suggests the occurrence of multiple phases of wedge growth, with a recent wedge growth activity occurring during the Holocene-Pleistocene boundary. The latter is derived from a minimum age model (Zinelabedin et al., 2022). The timing of the last wedge growth activity described by the authors would appear to coincide with the Central Andean Pluvial Event (CAPE) at 13.8–8.5 ka (CAPE II; de Porras et al., 2017). This

period would potentially provide sufficient moisture to (re-)activate haloturbation processes and wedge formation. However, Zinelabedin et al. (2022) concluded that the wedge stratigraphy remains unresolved due to insufficient subsampling resolution requiring further research to determine the age of calcium sulphate wedge formation.

Given that the investigated outcrop is situated within the summer rain regime (Houston, 2006) that prevails at the Andean foreslope, sporadic rain events that occur in this area could provide sufficient moisture to feed

haloturbation processes and hence wedge formation in the subsurface. The distance of the outcrop from the Aroma fan site to the coast and its altitude (outcrop located at ~1630 m a.s.l.) suggest that fog advection is an

unlikely mechanism for moisture supply at the Andean foreslope (fog advecting from the Pacific is mainly restricted to altitudes <1200 m a.s.l., Cereceda et al. 2008). Based on plant-specific *n*-alkane data from surface sediments and soil profiles, Mörchen et al. (2021) found that the Aroma fan region was affected by rain rather than fog. The authors thus concluded that episodes of higher water availability (and vegetation) had previously occurred at the Aroma site. Given the proximity of the Aroma fan outcrop to the winter/summer rain boundary (see Fig.1; isohyets based on Houston 2006), the outcrop is more sensitive to variations in winter/summer rain. Therefore, subtle changes in winter and summer rain could result in significantly more rainfall at this site. The presence of Al-bearing sulphates such as aluminite and celestine (Sr-bearing sulphate) also suggests that sufficient moisture was or is currently available to initiate the leaching of Al and Sr from the minerals. However, this moisture level is not sufficient to remove large quantities of calcium sulphate from the deposits. The dominance of calcium sulphate in the Aroma fan deposits indicates that mean annual precipitation is unlikely to have exceeded ~30 mm/year (Rech et al. 2003; 2019).

Polygonal patterned grounds can also be observed on surfaces at the northern and southern rims of the Río Loa Canyon situated within the hyperarid core of the Atacama Desert (see Fig. S.11, cf. Allmendinger and González, 2010; Mohren et al., 2020). It remains unconfirmed whether wedge polygon formation processes are currently active in this locality. However, the local influence of fog (e.g., Cereceda et al., 2008; Schween et al., 2020) and the presence of gypsum crusts (Mohren et al., 2020) suggest the potential for episodic salt-induced wedge-polygon formation in this region. Thus, a comparison of wedge-polygon structures from different sites in the Atacama Desert is essential to constrain their formation conditions. Furthermore, an understanding and timing of wedge-polygon-formation under hyperarid conditions may also be important for interpreting wedge-polygon formation in other water-limited environments, such as on Mars (see Fig. S.11). The correlation between polygonal patterned grounds and ground ice on Mars (e.g. Mangold, 2005), has led to the interpretation of their formation mechanisms as periglacial wedge-polygon formation, as described in numerous previous studies (e.g. Mangold et al. 2004; Mangold, 2005; Balme and Gallagher, 2009; Levy et al., 2009, 2010; Hauber et al., 2011; Soare et al., 2014). The presence of salt minerals on Mars (e.g. Clark and Van Hart, 1981; Osterloo et al., 2008; Hanley et al., 2012; Bishop, et al., 2014; Ehlmann and Edwards, 2014; Vaniman et al., 2018) and in particular hydrous sulphates (e.g. Gendrin et al., 2005; Dang et al., 2020; Rapin et al., 2023) suggests that salt-induced swelling and shrinking due to hydration and dehydration or thermal contraction could be additional potential mechanisms for the formation of polygonal patterned ground in regions with limited ground ice on Mars (less than 6% ground ice mass; Mangold, 2005).

**6 Conclusion**

It is hypothesised that the subsurface wedge and soil crack network of the Aroma fan in the northern Atacama Desert was predominantly formed by haloturbation processes, which were dominated by swelling and shrinking processes of calcium sulphate phases. Given that haloturbation and subsurface wedge formation require meteoric moisture, wedge formation is likely to have occurred under wetter but still (hyper-)arid climatic conditions. The surface crust is most likely the product of long-term net atmospheric deposition of calcium sulphate dust. Surface exposure ages ($^{21}$Ne) obtained from clasts situated on top of the crust date back to the Pliocene, indicating that the clasts were lifted by the accretionary dust mantle. The long-term accumulation of salts requires a hyperarid climate, which is why we interpret the surface crust to have formed under drier climatic conditions than those prevailing when the wedges were formed. Due to this environmental change, we suggest that subsurface wedge and polygonal patterned ground formation may have been attenuated or stopped. Although soil fines continue to be relocated downward within the surface crust, the majority of sediments and moisture may be retained at the surface in the long term. Further age information from the wedge and crust material is required in order to resolve the timing of the haloturbation processes and the climatic shift towards more hyperarid conditions (crust formation). A comprehensive understanding of wedge-polygon formation by haloturbation under hyperarid conditions could serve to complement existing hypotheses regarding wedge-polygon formation in other water-limited or hyperarid environments, such as those observed on Mars.

**Author contributions**

TJD, BR, and AZ conceptualized the study. The project was supervised by TJD and BR. Sample preparation and analyses were performed by AZ. X-ray diffraction measurements were carried out by MWW. Photogrammetry was conducted by JM and AZ. JM performed the preparation of the plutonium samples. SH was responsible for the plutonium measurements. The manuscript was drafted by AZ and internally revised by all authors.

**Conflict of Interest**

The authors declare that they have no conflict of interest.

**Data Availability Statement**

All data generated during this study are included in this published article and its supplementary material.

## Acknowledgements

This project is affiliated to the Collaborative Research Centre (CRC) 1211 "Earth – Evolution at the Dry Limit" (Grant-No.: 268236062) funded by the German Research Foundation (Deutsche Forschungsgemeinschaft, DFG), Germany. We would like to thank Hanna Cieszynski (University of Cologne) for support with the SEM measurements. We thank Nicole Mantke (University of Cologne) for the performance of the grain size analysis and Jochen Scheld (University of Cologne) for the assistance with the ICP-OES analysis. We would like to thank Olympia Nita (University of Cologne) for crushing the XRD samples. Finally, we would like to thank Eduardo Campos and colleagues at the Universidad Católica del Norte in Antofagasta for their logistical assistance during the field campaigns. We thank the reviewers Rui-Lin Cheng and Christof Sager for their constructive feedback, which improved our manuscript.

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
