# Peer review of "Haloturbation in the northern Atacama Desert revealed by a hidden subsurface network of calcium sulphate wedges"

_EGUsphere, 2024_

## Referee Comment (RC1)

Reviewer: Rui-Lin Cheng, The University of Hong Kong

**Overview**

The manuscript submitted by Zinelabedin et al. (manuscript No.: egusphere-2024-592) presents a comprehensive investigation of the interesting and unique salt wedges hidden in the subsurface in the northern Atacama Desert, using a variety of analytical methods to examine both surface salt crust and subsurface salt wedges. The results indicate that haloturbation is the primary process that has formed the salt wedges and inferred polygonal patterned ground. This study also links surface/subsurface processes to the changes in the climate and interactions with the atmosphere within the temporal constraints of surface exposure dating. Overall, this is a well-written manuscript and represents a useful contribution to the community.

However, I have major comments regarding clarifying the formation processes and the extrapolation of this work to Mars, as well as a few minor suggestions for improving clarity in certain areas. Please find details in the attachment. I would recommend it for publication with the condition that moderate revisions are made to address the comments I have provided.

**Major comments:**

1. Further clarification is needed regarding the formation of salt wedges and polygons in this study.
    a. The three main proposed formation processes of salt wedges and polygons are haloturbation, thermal contraction, and desiccation, which have been mentioned throughout the text (i.e., lines 51-55, 111-114, and 346-348).
        i. The introduction part lacks a brief overview of the latter two processes.
        ii. The exclusion of the latter two processes from this study requires more justification. This has been briefly mentioned in lines 344-348. However, a detailed discussion about the differences between the features observed in this study and those dominated by thermal contraction or desiccation in Atacama would be helpful.
    b. To enhance clarity, it is better to provide specific descriptions of salt minerals.
        i. It is recommended to use specific sulphate terms such as "anhydrite" and "gypsums" instead of "calcium sulfate" whenever possible to facilitate reader comprehension of the discussed salts. For example, in lines 376-380, which type of salts caused the volume increase? In lines 398-400, which salts caused the crack opening, and which salts/materials caused the filling?
        ii. It is worth considering the estimation of volume changes caused by sulfates in the study, similar to the calculation presented in lines 60-65 and 410-415. This analysis could provide additional insights into the impact of sulfates on the formation processes.

2. The extrapolation of this work to Mars
   The Atacama Desert is a good terrestrial analog for Mars. And it is reasonable to extend the findings of this study to Mars and share them with a broader scientific community. However, the content presented in lines 103-106 and 552-562 requires additional information to support the comparison/analogy.
   a. Line 104: "concluded" -> "suggest"
   b. Line 557: Osterloo et al. (2008) did not interpret polygonal morphologies related to a periglacial origin.
   c. Lines 559-560: While this study focuses on Ca-sulphates, the references listed here include a diverse range of salt minerals.
      i. In this study, phase transitions among different sulfate phases can cause volume change. Does this mechanism apply to other salts (e.g., chlorides, chlorates, and perchlorates)?
      ii. Among the references cited, only Dang et al. (2020) reported polygonal morphologies on Mars, with the interpretation involving desiccation. How can formation processes proposed in this study contribute to a better understanding of Martian polygons?
   d. Recommended references:
      - Rapin et al., 2023, Nature, https://doi.org/10.1038/s41586-023-06220-3
        This work reported Ca/Mg- sulfate-enriched polygonal ridges at Gale Crater, Mars.
      - Cheng et al., 2021, Geomorph., https://doi.org/10.1016/j.geomorph.2021.107695
        This study reported polygonal features controlled by Ca/Na-sulfates at the Qaidam Basin and discussed the implications for Martian polygons.
      - The enrichment of hydrous sulfates (gypsum and aluminite) without halite in the surface crust at the studied site (Section 5.2 and line 510) is quite interesting to me. The presence and distribution of hydrous sulfates on the surface of Mars have intrigued the community (Gendrin et al., 2005, Science: https://doi.org/10.1126/science.1109087 and references therein). Extrapolating this study to Mars from this perspective may provide valuable insights. Nevertheless, this might deviate from the original focus. Thus, I leave it here for open discussion.
      - (following the point above) Zhu et al., 2024, Geomorph., https://doi.org/10.1016/j.geomorph.2023.108934
        This study reported halite enrichment on the surface of polygons in the arid environment in the Qaidam Basin. It is also intriguing to explore the factors that may cause such differences in mineral enrichment.

**Minor comments:**

Lines 43 & 103: The missing reference of Amundon (2018) in the bibliography list. Please also check other references.

Lines 57-59: Need some rewording.

Line 66: At the beginning of a paragraph, "such processes" refers to?

Lines 66-96: It seems like this part can be briefer in the introduction. Some can be moved to the section about the regional setting; other can be moved to the discussion regarding the moisture source/input.

Line 103: Additional references should be included.

Fig. 1A: Add the legend for the red squares or mention them in the captions. The sketch in Fig. 1E is excellent.

Lines 148-151: Fig. 2 seems more like results rather than serving as the illustration of the collected samples. Fig. 3 can be moved to Section 3 to illustrate the sampling locations. Add in-text Fig. citations to enable easy navigation to photos of the samples.

Line 234: Are these soil cracks related to desiccation? Please also clarify the terms: "soil" vs "salts" vs "samples", and "soil cracks" vs "salt wedges", throughout the text.

Line 268, 273-276, and Fig. 4: It is recommended to directly use "wedge center" and "periphery" (and label them in Fig. 4B) instead of "LP" and "RP."

Line 306 & 336: It would be more concise to mention "aluminite" specifically instead of using the phrase "evaporite content (except for Ca-sulfate).

Line 316: The dimension of the crack size should be specified.

Fig. 5: Add a scale bar for the sample.

Line 381: "Formed by subsurface pressure" is confusing.

Fig. 6: The red arrows only illustrate the swelling stress. How about the shrinking one?

Fig. 7: Add the scale bar for all images. What materials, particularly the type of salts, are present in each site?

---

## Referee Comment (RC2)

Reviewer: Christof Sager, Museum für Naturkunde - Leibniz Institute for Evolution and Biodiversity Science; 10115 Berlin, Germany.

*1. An initial paragraph or section evaluating the overall quality of the preprint ("general comments")*

Zinelabedin *et al.* (manuscript No.: egusphere-2024-592) present a compelling study on polygonal patterned ground featuring sulphate-cemented wedges overlain by a sulphate crust in the Atacama Desert. This research employs a robust multi-method approach to investigate these unique features, which stand out from previously studied polygon-wedge systems in the region.

The authors attribute the formation of these wedges primarily to so-called haloturbation processes and the expansion-contraction dynamics (swell-shrinking) of calcium sulphate under hyper-arid conditions with sporadic rainfall events. The study is relevant for the readers of Earth Surface Dynamics as it makes a significant contribution to the research field of arid environments, while particularly enriching our understanding of patterned ground. Furthermore, its implications extend to extraterrestrial studies, particularly in understanding patterned grounds on Mars. Overall, the study is of good scientific quality and is well written, with the data presented appropriately.

However, there are several aspects, including some major concerns, that need to be clarified before the manuscript may be published. These concerns do not relate to the presented methods or acquired data, but rather to the interpretation of the results and the terminology used for salt-related processes and wedge formation. Addressing these issues should not pose significant challenges and will enhance the manuscript's clarity.

*Major comments:*

A major concern is the use of the term "haloturbation" as the dominant process for polygon-wedge formation, which I believe is not appropriate. First, this process is not explained in sufficient detail and is not clearly differentiated from other salt-related processes forming polygons/wedges, potentially leading to confusion for the reader. Second, the authors attribute the wedge formation in lines 413-415 mainly to shrinking processes due to phase transitions of sulfates, rather than directly to haloturbation, "which dominates in the polygon body, causing salt heave.". In my opinion, shrinking-swelling should not be seen equivalent to or be summarized under the term haloturbation (as in line 396), as shrinking-swelling is rather a term for volumetric changes in sediment or soil. Thus, shrinking process (or more general contraction) as the authors state in lines 413-415 is more appropriate than haloturbation as the main driving force for repeated soil cracking and thus, wedge formation. The authors already present very strong indication of hydration and dehydration processes by the presence of different calcium sulfate hydration forms with the XRD data. In contrast, haloturbation refers to the deformation of the original soil or sediment texture by the precipitation or dissolution of salts, but it remains too vague how it leads to meter-deep soil cracks. Therefore, the argument in line 381 that soil cracks are formed by subsurface pressure is questionable without further elaboration and contradicts the statement in lines 413-415 where contraction (shrinking) rather than expansion (which leads to subsurface pressure) is attributed to soil cracking.

For easier understanding, I suggest the following terms be clearly defined and differentiated in the introduction section: salt heave, clast/salt shattering as a form of salt weathering, contraction and expansion in the context of shrinking-swelling, thermal contraction, dehydration and hydration of minerals, and desiccation (as a crack formation mechanism). The term haloturbation is not necessarily needed when considering the above-mentioned concerns. It seems that these terms are not consistently defined throughout the literature, and their use and definitions vary across disciplines and studies which can lead to confusion for readers not familiar with these processes.

My understanding of the terms: contraction and expansion are more general terms used for volumetric changes in sediments and soils. Shrinking and swelling are commonly used for volume changes due to

changes in water content in clay minerals with swelling potential, occurring in e.g., in playa environments where rain events lead to initial swelling of clays and subsequent drying/shrinking, resulting in desiccation crack polygons. For phase transitions between gypsum, bassanite, and anhydrite, the terms hydration and dehydration (as a form of desiccation) would be more appropriate, distinguishing them from the shrinking-swelling of clay minerals. Therefore, I suggest the dominant wedge formation processes be termed contraction due to dehydration of calcium sulfates rather than using the term haloturbation.

A second concern is the interpretation of wedge cementation. Did the authors consider alternative explanations? The authors correctly state that in other studies, the wedges were largely free of salts or salt-poor. In Sager et al. (2023) (https://doi.org/10.1029/2022JG007328), a rain experiment was conducted showing that salt precipitation occurs on the surface of the wedges after wetting, likely due to upward movement of saline water along the wedges. Additionally, Sager et al. (2021) observed that the outer parts of the wedges had higher salt content than the inner parts. It was suggested that, over time and with sufficient rain events, salts migrate from salt-rich polygons towards the initially salt-poor wedges, eventually cementing them. Therefore, it is very interesting that Zinelabedin *et al.* observed so intensely cemented wedges. In Figure S.1 it appears that the wedge center shows a lower sulfate content (brownish colors) compared to the periphery (whitish color). The processes proposed by Sager *et al.* could also be relevant for the cementation of the Aroma fan wedges. Can the authors discuss this possibility, or can it be excluded that the calcium sulfates in the wedges originated from the polygon? Addressing these alternative explanations would provide a more comprehensive understanding of the wedge cementation process and strengthen the study's conclusions.

**2. Section addressing individual scientific questions/issues ("specific comments") and technical corrections**

**Title & Abstract:** The title and abstract should be changed regarding the major concern that haloturbation might not be the appropriate term. The summary would benefit from more information, e.g., what is the composition of the polygons (the sediment/soil) between the wedges, how deep are the wedges and how many wedges were examined? The authors were able to determine an age for the crust. Since such ages are very important and rare, I would add them to the abstract.

**Lines 11-12**
Consider rephrasing "post sedimentary features", since it remains questionable if wedges are completely post sedimentary features since they are formed by sediment deposition in the cracks and are accompanied by ongoing sedimentary processes as dry dust and salt deposition migration by percolating rainwater. Or rephrase, if with post sedimentary is meant after the deposition of the host material.

**14-16**

From my point, these are very interesting polygon wedges, due to the high anhydrite content. However, wording suggests that the high anhydrite content lacking in other locations implies different formation process. However, it could be also post-formational processes that led to these type of wedges. As stated above in the major concerns, I would add the possibility (maybe better in the discussion) that overprinting of sand wedges by calcium sulfate from the polygon could occur as a secondary process after initial wedge formation.

**16-17**

Please rephrase under consideration of major concerns. Also, it is unclear who assume the process, the authors or the general scientific community. Better: "We assume contraction/shrinking due to dehydration of calcium sulfates to be the main driver wedge formation at the Aroma fan site."

**23-24**

Which are the other processes, do you include thermal contraction and desiccation as possible processes?

**Introduction**

**35-38**
I would disagree that cryoturbation is the main mechanism for (non-sorted) polygon formation. Cracking and thus wedge formation is rather caused by stresses from rapid cooling or low temperature (thermal contraction), see Lachenbruch (1962) (https://doi.org/10.1130/SPE70). After initial cracking, seasonal freeze-thaw cycles allow water to enter the cracks and subsequently freeze, which forms wedges over time. However, sorted patterned ground, such as sorted stone circles (which do not have classical wedges as non-sorted patterns) are assumed to be a result of repeated frost heave processes (e.g., Kessler 2001, https://doi.org/10.1029/2001JB000279). Thus cryoturbation is a process often occurring in the periglacial soils, but do not necessarily contribute directly to the pattern formation.

**38-39**
This is correct, maybe the authors could elaborate when wedges are expected and when not. The differentiation between sorted and non-sorted patterns could help, as all non-sorted polygonal ground is defined by the presence of wedges (filled cracks) or unfilled cracks, that separate the polygons.

**43-44**
Please rephrase the sentence, as "strongly differing" relates probably to the periglacial environments, but it reads as if this comparison is drawn to Mars, since it was discussed above.

**46**
"wedge structures can also be found in the "Atacama Desert" instead of "here".

**52**
Buck et al. 2006 is very interesting but unfortunately only a conference abstract. Do the authors have found the appropriate manuscript to this study? Also, Buck et al. write:
"These features are interpreted to have formed through salt heave, which occurs when salt minerals cement soil grains creating the cohesion necessary for tensional stresses (caused by desiccation, and/or thermal contraction of salt minerals) to form contraction cracks. The contraction cracks are filled with eolian dust (salt/sediment), preventing their closure during periods of expansion caused by salt mineral precipitation and/or thermal expansion." Which means that wedges and cracks are formed by contraction rather than the salt heave itself.
In Ewing et al. 2006 it is said:
"The presence of sulfate polygonal prisms in multiple horizons (Fig. 10; Table 5c) suggests cyclical hydration and dehydration of gypsum/anhydrite (Chatterji and Jeffrey, 1963)" please consider putting this reference behind "dry environments" rather than "haloturbation". Or exchange haloturbation with contraction/dehydration.

**60**
What is was/where the other process or processes in addition to gypsification?

**66-67**
Unclear wording, please rephrase. Do the authors want to say the dehydration /hydration processes were observed or potentially can be identified?

**2 Regional setting**

**126-128**
Regarding wedge depth, consider writing "between 1-2 m", since in Fig. 1 you state that cracks are all >1 m

**128-131**
Consider adding these sentences to the results section rather than the regional setting section.

**Figure 1**
General:
Please homogenize the size of frames and labels, e.g., frame of panel C is smaller than E frame.

Panel A: Please refer to the map of South America and the red dot in the lower left corner. Consider using a frame.
Please give numbers for all isohyets, as it is missing for the one above 200 mm/yr.
Consider combining the symbols for wedge and polygon structures as except for dessication crack polygons, wedges and polygons are one system and it would simplify the map and enhance the position of published structures.
There may be an unintended black line at the lower end of the 100mm/yr line and below "Gonzalez" further to the left.
Is there a reason for the non-continuous coloring style of the DEM (orange vs gray, 2000m vs 2500 m)? If appropriate, consider using a continuous shading.
PANEL C:
"Displaying a subsurface network of large soil cracks (>1 m depth) and vertically laminated wedge structures"
It could appear to the reader that soil cracks and wedges at the outcrop are two distinct features appearing next to each other, but in this case the belong together (I admit that in general cracks can be present without wedges and vice versa). Consider rewriting the sentence. e.g., "vertically laminated wedges with vertical soil cracks along their centers".
Panel D:
If D is a close-up can the authors show the positions in C?
Panel E:
This is a nicely drawn sketch. I wonder is there a reason for the more whitish colors in the upper part, and the more brownish colors in the lower part of the polygonbody, left of the '5'. The same for the surface crust on the right side of the sketch, which seems to be underlain by a darker surface crust? Please clarify.
**142** „Detailed description of the wedge network is given in the result section" → I think this is not necessary and can be deleted.

**3 Material and methods**

**148**
Consider using just 'trench wall' as in Fig. 1 instead of trench sidewall

**148-151**
It is unclear how many samples were collected and at which depths. E.g., What were the dimensions, weight and depth of the surface crust block? How many samples were taken below the crust, e.g., from wedges or polygon?

**152**
What kind of foil was used?

**154**
The multi-methodological approach was only applied to the crust and wedges but not to the polygon? Was there a reason for this?

**159**

Considering the high sulfate cementation, can the authors elaborate how was sampled, using a spoon, hammer, jackhammer?

**164**
Please add the depth of the base of the crust.

**165**
Can the authors elaborate how the powder samples were generated? E.g.; hand grinded, grinding mill, grinding duration, temperature during grinding, drying of samples etc., also to ensure for the reader that a phase transition occurred accidentally during sample preparation.

**167-169**
If whole powder pattern fitting was used for 'Quantitative phase analysis', I recommend adding this information to the sentence.

**176**
EDX stands for energy dispersive X-ray spectroscopy.

**179-181**
How did the authors verify that all the $CaSO_4$ was removed from the sediment? In Figure 4, largest grain size is coarse sand, what happened with the fraction larger 2 mm, was it excluded? If so, I would add this information.

**183**
"Etched" suggests that not all carbonates were removed. I suggest to use another word for HCl treatment. Delete space between '10' and '%'.

**185**
It seems that ICP-OES is used here for the first time. When used for the first it should be spelled in full.

**187**
Please summarize the procedure by Voigt et al. shortly.

**200**
How was sieved, wet or dry, by hand or sieving machine?

**209-211**
Please rephrase to "we sampled dust/sediment from a cavity" instead of "we sampled a cavity".

**4 Results**

**231-233**

I would assume that the sulphate-cemented sediment contains also clasts smaller than pebbles. If the authors want to highlight that pebbles to boulder sized clasts are visible in a more fine-grained matrix, the sentence should be rephrased. If referring to 'Figure3 panel c', panel a and b must be mentioned first, or at least the whole figure 3.

**233-234**
From figure s3 it seems that ~30% of crack filling are other salts than anhydrite, can the others state which are these salts or add here the word "mainly" or "dominantly"? It is not clear if the authors refer to table S3 or Figure S3 → '(see S.3 in the supplementary material).'

**240 Figure 2**
I think the combination of a photograph in the background with data presentation in the foreground is

not necessary. Consider just presenting the data in figure 2, since the aroma fan surface is shown in Fig.1. alternatively add the oblique background image of the fan surface to Figure 1. If the whole photograph is directed into NE direction, I suggest to remove the NE notation in the top right corner as it is not appropriate.

**245 Figure 3**
Legend: 'Macroscopic' instead of 'macroskopic'
Panel notation a) and b) is both at the Aroma site, why is it only notated at panel b? I suggest to remove 'Aroma site' at panel b or to add it at panel a as well.

**255**
The detailed investigation of the calcium sulfate polymorphs is very interesting. Can the authors elaborate how this polymorph of anhydrite was identified or differentiated from other polymorphs, e.g.; which database was used or provide diffractograms etc.?

**257**
Please add the formula of halite as for the other evaporites.

**264-265**
Please reference to the corresponding data or figure at the end of the sentence.

**265-267**
I think this is very important observation, one could add, that this is visible by the color change of the wedges as well. The increasing cementation towards the periphery was also observed for the polygons in the Yungay region by Sager et al. 2021: "*Furthermore, the cementation of the marginal and thus older parts of the SW indicates that rain events have occurred since the SW formation and the soluble content is not only leached downward but also migrated in horizontal directions. Although the vesicular horizon has formed in the margin of the SW, this cementation must take place without an intense deformation because the vertical lamination is still visible (Fig. 4)*".
However, the next sentence in **lines 267-269** is confusing as it states the opposite, if I understood correctly? Please clarify.

**272**
Instead of 'match' use 'correlate', and state, if possible, the $R^2$ value.

**281**
Please add formula of aluminite.

**286 Figure 4**
Panel B: The white dots indicate XRD sampling positions. In the supplementary material in *Figure S.6, the* XRD data is given for sample *ARO18-02-001. Do I understand it correctly that this data corresponds to the most right white dote with the number 1. If not, please clarify.*
Panel B.1 it is hard to see, where the arrows for e.g., gypsum or anhydrite are pointing at. Could the authors highlight the clasts or part of the cement somehow, so that this becomes clearer?
Panel B.2: How were the wt% calculated, or is the data maybe semi-quantitative, then it should be [%].
Panel B.3 In the grain size diagram, it appears that a frame around the individual size fraction columns was used, leading to a poor recognizability (on my screen) of the small size fractions. If the fraction would be displayed without a frame, the visibility of the smaller size fractions could be increased. The authors also mention pebbles in cobbles in the sediment, why is the largest grain size fraction coarse sand? Was the other material excluded?

**297**
"crust surface" instead of "crust top surface" as surface should be always 'top'.

**304-306**

Regarding the XRD data of crust surface, please provide sampling depth and refer to the corresponding data or figure at the end of the sentence.

**306-308**
'Subsamples taken from a few centimetres below the surface' → please provide sampling depth

**312-314**
Please refer to the corresponding XRD data or figure at the end of the sentence.

**316-317**
Are the >15 cm the cracking width or depth?

**355 Figure 5**

Figure 5 is a nice figure with a lot of information. But with the current organization, it can be overwhelming for the reader. I suggest reduce the presented information and organize the figure into panels with frames and/or labels. This will also increase readability of the figure caption. In Figure 4 the individual SEM images are not labelled, while here in figure 5 each image was labeled (SEM 1, SEM 2 etc.). In Figure 4 the grain size dots are grey, while in figure 5 they are orange. The XRD data is present without [wt%] but not in figure 4. Please use the same layout for both figures.
The colored images right of SEM 3 are not explained in the figure caption, it appears that they belong to SEM 3, consider providing scale for these images as well. Also, SEM 3 appears two times, mark one as a zoom-in using frames, if this is the case.
Please explain all abbreviations also in the figure caption, e.g.; GS = grainsize
The information of the 'general crust data' should be presented in the text instead of in the figure, to further clear the Figure.

**Discussion**

**General comments:**

**Please revise the discussion regarding the major comments (haloturbation vs dehydration of calcium sulfates/shrinking)**

**344-348**
In the context of the major concerns, it remains unclear why the presence of Calcium sulfates in the wedges is an indication for haloturbation as dominate wedge formation process. As the authors nicely investigate different calcium sulfate phase and polymorphs it would be appropriate to state the mineral phases instead of the general term "calcium sulfate". As mentioned further above, cemented wedges were also observed in Yungay, but with a lower cementation degree, and a cementation of the wedges that corresponded to the cementation in the polygon, e.g., gypsum at 10 cm, anhydrite at 30 cm and halite at 70 cm depth. Consider adding a short paragraph comparing your results with wedge results from other studies in more detail, as this would be very interesting.

**352**
It is unclear what is meant with 'pattern' here?

**355-358**
Although the formation of anhydrite polymorph is still under debate, can the authors imagine or disprefer dehydration/shrinking processes that leads directly to β-anhydrite or would gypsum dehydration inevitable lead to γ-anhydrite that needs to be dissolved in a highly saline solution and the re-precipitated as β-anhydrite? In this sense it would be also interesting what anhydrite polymorph is dominating in the polygon. If one would assume that polygonal cracks form by dehydration, the polygon should contain γ-anhydrite (as a result of dehydration) and if the wedges are cemented over time by saline solutions, one could expect β-anhydrite. Another question would be if β-anhydrite can

hydrate back to gypsum, do the author have references or hypothesis to this? β-anhydrite is often considered insoluble in the literature, do the authors know if this is also the case for geological timescales? Although it would speculative, the authors could give more insights on their thoughts regarding this topic.

**358-361**
How would primary gypsum be differentiated from secondary gypsum?

**362**
"silicon and phosphorous". Earlier in the manuscript, the authors use the element symbols (e.g., for S and Ca) instead of element names. Please homogenize.

**364-367**
This is very interesting point, as it could explain dehydration (shrinking) processes for both anhydrite polymorphs γ-anhydrite and β-anhydrite.

**368-415**

Please edit this section regarding the main concern of haloturbation vs dehydration (shrinking). Also why does the saline solution exceeds saturation at greater depth and at which depth exactly?

**378-381**
This is a critical sentence that should be rephrased or deleted, as there is no reference or argument for crack formation by subsurface pressure, further it contradicts the shrinking argument made by the authors in lines 413-415. The argument that expansion leads to upward deformation may be correct, but it is no explanation for soil cracking.

**386-387**
Consider elaborating here about cementation of wedges by repeated rain events as mentioned above.

**387-391**
Consider adding that swelling shrinking applies probably mainly to the polygon. It could also be added that saline water migrates also through pore space and not only along the crack.

**392-394**
How do the authors know that salt heave processes intensify and do not occur e.g., with the same magnitude over time? I agree that salt heave mediated microtopography can visualize a polygonal surface pattern, but it is not the only reason why a pattern can be observed. In the Atacama Desert, the patterns are also visible due to color differences, e.g., darker or brighter wedges compared to the polygon body as a result of compositional differences or by morphology, e.g., high-center and low-center polygons. Morphology differences can be a result of erosion or soil deforming processes, including thermal expansion or salt heave (as the authors state correctly) leading to elevated polygon shoulders (low-center polygons). Therefore, please elaborate on the visualization of polygon patterns in the field.

**403-406**
I agree that the outer parts of the wedges are likely the oldest part and I support the idea that salt-related processes may destroy initial lamination. But I disagree that wedge formation is possible without initial soil cracking. Therefore, I suggest to elaborate on this or to remove the statement.

**408-410**
Do the authors mean phase transition from gypsum to anhydrite at the center or periphery, please clarify?

**410-415**
Again, I really support the attribution of crack opening to shrinking, but it should be differentiated from haloturbation.

**Figure 6**

Figure 6 is a nicely drawn sketch, but considering my main concern, I suggest some re-editing.
Panel 1: looks nice, no comments.
Panel 2: I would show only the precipitation of salts and if wanted some haloturbation after the rain event (step 2). As a result of haloturbation also the original texture should change, leading to dislocation of some of the drawn clasts or some salt heave of clasts at the surface. I would move 'shrinking (dehydration)' and 'crack formation' and put it in the third panel (step 3), as this should occur after step 2. The red arrows that lead to the crack are due to contraction forces. However, this could be only visualized if the authors would show two wedges and a polygon in the center, where the arrows are directed towards the polygon center. As it is now in panel 2 shrinking and swelling seem to be directed in the same orientation.
Also in Panel 3, the subsequent filling of the cracks by aeolian-derived sediment, leading to the first vertical lamina should be visualized.
In Panel 4, I would show that the repetition of step 1-3 leads to wedges and clast shattering in the polygon over time. Then the formation of the surface crust can follow. If the authors do not recognize a microtopographic signature by salt heave I suggest to not draw it (or at least show it less intense).

**437-476**
The section regarding the crust formation is very interesting. Could the authors maybe elaborate on the extent of the crust. Is it only locally or covering the whole aroma fan, and is the crust also present in e.g., old gullies or river channels? How do the authors know the aroma fan surface is free of a surficial polygonal patterned ground, was there some reconnaissance by drone imaging or by satellite imaging (e.g., Google Earth)?

**480-482**
Please refer to the corresponding data or figure.

**488**
Aluminite is probably not detected in the clast, but rather in the filled fractures of the clasts right? "Aluminite is also detected in the fractured clasts in…" change to "Aluminite is also detected in the fractures of the clasts in…"

**498-500**
Can the authors elaborate why "presence of sulphates reflects minor dissolution and reprecipitation of salts". Why only minor, please clarify.

**504**
"as β-anhydrite also requires highly saline solutions to precipitate." Maybe the authors want to add "presumably" as they state earlier that formation of anhydrite polymorphs is still under debate.

**550-551**
This seems like a completely new model of polygon-wedge formation by fog. Is this idea presented by Cereceda et al. or by Schween et al.? If so, the reference should be at the end of the sentence. If it is a hypothesis of the authors, they should either explain it in more detail or consider removing it, as this would be an interesting topic but may not be within the scope of this study.

**560**
Instead of haloturbation the authors could discuss the relevance of other salt-related processes for polygon formation on Mars (salt based thermal contraction/expansion or dehydration/hydration).

**Figure 7**:

Figure 7 is a nice figure showing polygonal ground from different sites with different conditions, but I think the information content is too low to show it in the main manuscript, also since the comparison of polygonal ground between periglacial areas, Atacama Desert and Mars in the text is rather short, and that the polygons near Rio Loa were not investigated in this study. I suggest to remove the figure or to place it in the supplementary material.

**Conclusion**:
The conclusion can be shortened, as it reads more like a summary rather than providing new concluding remarks. E.g.; it could be highlighted that with this study another type of polygonal patterned ground has been identified and described in the Atacama Desert, which broadens the diversity of patterned grounds in the Atacama. Further, the analysis of the crust indicates that polygonal ground (usually a near surface feature) can be buried, thus enabling the possibility of multiple generations of polygonal patterned ground along a vertical ground profile. To make a link to Mars, the authors could mention the recent identification of buried polygonal ground underneath Utopia Planitia (https://doi.org/10.1038/s41550-023-02117-3). Given the new age constraints of the crust the authors could also highlight that polygonal patterned ground can be rather old/ancient landscape features in the study region (e.g., in contrast to recent gullies or fluvial deposits after the rain events from 2017).

**845**
The doi link seems not to be working for the reference.

**Supplementary material**

Figure S.1
Can the authors provide a scale for the photograph?

Figure S.2
Can the authors provide a rough size of the board used as scale for the photograph?

Figure S.3:
Please add more information to the figure description, e.g., that XRD data is presented, consider using panels to organize the figure. Where are the two black lines pointing to? Is the XRD quantitative or semi-quantitative please add this information too.

Figure S.4:
Do the authors maybe wanted to say "red points" instead of black points, as I don't see black points.

Table S.4: XRD results
Can the authors state the unit for XRD analysis, e.g.; % or wt%?
For the more exotic minerals such as Aluminite or Konyaite, could the authors provide individual diffractograms as done for Figure S.6. The authors could also highlight the main peaks for these minerals.

Figure S.6:
Can the authors add the information that the sample belong to the surface crust.

Figure S.7:
The illustration description reads redundantly, please shorten it.

Figure S.9:
Could the authors leave a "white space" between the individual figures, as is the case with Figure S8?

Photogrammetry/Methology:
What is meant with: "The sample was turned upside down and pictured in four runs" where it the four

runs with similar camera orientation or was the camera angle changed for each run?

If the picture count is stated for ARO18-02, it should be done for the other samples as well (or leave it out here, as it is stated in the Agisoft reports)

Why was the scaling of the 3D model done with distinct features of the sample when scale bars were laid out, or was it the combination of both, please state?

How was the density of ARO18-02 determined using photogrammetry if it could not be turned around?

For final publication, authors should consider presenting the Agisoft processing reports as a separate file, as page numbers, figure and table comments do not match the rest of the supplementary material, or editing the reports, which is likely to be more time-consuming.

---

## Author Response (AR1)

**Responses to review comments on manuscript EGUSPHERE-2024-592**

**Reviewer #1 (Rui-Lin Cheng):**

*R(1): Overview*

*The manuscript submitted by Zinelabedin et al. (manuscript No.: egusphere-2024-592) presents a comprehensive investigation of the interesting and unique salt wedges hidden in the subsurface in the northern Atacama Desert, using a variety of analytical methods to examine both surface salt crust and subsurface salt wedges. The results indicate that haloturbation is the primary process that has formed the salt wedges and inferred polygonal patterned ground. This study also links surface/subsurface processes to the changes in the climate and interactions with the atmosphere within the temporal constraints of surface exposure dating. Overall, this is a well-written manuscript and represents a useful contribution to the community.*

*However, I have major comments regarding clarifying the formation processes and the extrapolation of this work to Mars, as well as a few minor suggestions for improving clarity in certain areas. Please find details in the attachment. I would recommend it for publication with the condition that moderate revisions are made to address the comments I have provided.*

Reply:  We thank the reviewer for the constructive and helpful comments on our manuscript. We have considered most of the comments and suggestions, which have led to improvements in the manuscript. Below, we provide a detailed, point-by-point response to each of the reviewer's comments (in black italic font), along with explanations of the revisions made (in blue normal font and new edited parts in blue italic font).

*R(1): Major comments:*

*1. Further clarification is needed regarding the formation of salt wedges and polygons in this study.*

*a. The three main proposed formation processes of salt wedges and polygons are haloturbation, thermal contraction, and desiccation, which have been mentioned throughout the text (i.e., lines 51-55, 111-114, and 346-348).*

*i. The introduction part lacks a brief overview of the latter two processes.*

Reply: We agree that the processes thermal contraction and desiccation need a brief explanation. We added a brief overview of the processes in line 65-68: *"Thermal contraction processes occur as a consequence of tensile stresses that develop in deposits during cooling (Lachenbruch, 1962). Desiccation cracking is, among other factors, caused by the dehydration of deposits due to the phase transition of hydrous calcium sulphate (gypsum; $CaSO_4 \cdot 2H_2O$) to anhydrous calcium sulphate (anhydrite; $CaSO_4$) (Cooke and Warren, 1973; Tucker, 1981)."*

*ii. The exclusion of the latter two processes from this study requires more justification. This has been briefly mentioned in lines 344-348. However, a detailed discussion about the differences between the features observed in this study and those dominated by thermal contraction or desiccation in Atacama would be helpful.*

Reply: We agree that a detailed discussion about the different features is needed. We have added the information that the Aroma fan wedge-polygon formation has developed on alluvial fan deposits, not in a playa environment. While the desiccation mechanism (involving the dehydration of calcium sulphate) in the Aroma wedges is similar to the desiccation polygons found in playa environments, the surrounding environment and deposits are different. Though we cannot entirely rule out thermal contraction processes in the wedge structures at the Aroma site, this mechanism is likely to play a minor role. The high calcium sulphate content in the Aroma wedges suggests that desiccation due to the dehydration of calcium sulphate is the more significant process driving wedge formation.

We have specified this in line 393-400: *"The Aroma fan wedge-polygon formation has developed in and on alluvial fan deposits, which contrasts with the polygon formation observed in a playa environment (e.g. Cheng et al., 2021; Zhu et al., 2024). While the Aroma fan wedges share a similar desiccation mechanism (calcium sulphate dehydration) with desiccation polygons in playa environments, the surrounding environment and deposits are distinct. Although thermal contraction processes cannot be entirely ruled out as a contributing factor in the formation of the wedge structures at the Aroma site, this mechanism is likely to play a minor role. The high calcium sulphate content in the Aroma fan wedges indicates that desiccation due to dehydration of gypsum is the most likely process responsible for the local wedge formation."*

*b. To enhance clarity, it is better to provide specific descriptions of salt minerals.*

*i. It is recommended to use specific sulphate terms such as "anhydrite" and "gypsums" instead of "calcium sulfate" whenever possible to facilitate reader comprehension of the discussed salts. For example, in lines 376-380, which type of salts caused the volume increase? In lines 398-400, which salts caused the crack opening, and which salts/materials caused the filling?*

Reply: We have specified the minerals "gypsum" and "anhydrite" instead of "salts" in line 449 (as both minerals can cause these processes). In line 475 we prefer the term "calcium sulphate" as the gypsum and anhydrite content vary between the individual wedges ARO17-03A and ARO18-08. In line 461-464 we have explained the crack formation based on calcium sulphate dynamics: *"Repeated cycles of frequent moisture events, or intermittent phases thereof, may have caused the accumulation of calcium sulphate within the soil crack. Swelling (hydration) and shrinking (dehydration) processes due to the phase transformation of gypsum*

*to γ-CaSO₄ and vice versa could have led to an increase in crack width and depth, as well as increased clast fracturing (see Fig. 6, step 3; cf. Howell, 2009)."*

*ii. It is worth considering the estimation of volume changes caused by sulfates in the study, similar to the calculation presented in lines 60-65 and 410-415. This analysis could provide additional insights into the impact of sulfates on the formation processes.*

Reply: We agree that the estimation of volume changes by sulphates is an interesting point. For calcium sulphate, volume changes are described in detail in previous publications (Milsch et al., 2011; Sanzeni et al., 2016; Butscher et al., 2017, 2018; Jarzyna et al. 2021), as we stated in line 87-92 and 488-490. It is not possible to accurately calculate any volume change due to the complex, heterogeneous, triangular wedge structures, which exhibit alternating vertical layer thicknesses and a reduction in thickness from top to bottom.

For the "exotic" sulphates e.g. aluminite and konyaite specific numbers for changes in volume are not directly mentioned in the literature.

*2. The extrapolation of this work to Mars*

*The Atacama Desert is a good terrestrial analog for Mars. And it is reasonable to extend the findings of this study to Mars and share them with a broader scientific community. However, the content presented in lines 103-106 and 552-562 requires additional information to support the comparison/analogy.*

*a. Line 104: "concluded" -> "suggest"*

Reply: Yes, we agree that the above-mentioned parts need more information to support the analogy. We have added the recommended literature from d) below in line 136-137 and 642. We have changed/rephrased "concluded" to "suggests" in line 133.

*b. Line 557: Osterloo et al. (2008) did not interpret polygonal morphologies related to a periglacial origin.*

Reply: Apologies for the mistake. Osterloo et al. (2008) should be cited in the following sentence, that salt minerals are present on Mars. We have corrected it in line 640.

*c. Lines 559-560: While this study focuses on Ca-sulphates, the references listed here include a diverse range of salt minerals.*

*i. In this study, phase transitions among different sulfate phases can cause volume change. Does this mechanism apply to other salts (e.g., chlorides, chlorates, and perchlorates)?*

Reply: Chlorides such as halite (NaCl) and sylvine (KCl) for instance, are not known to incorporate water into their crystal lattice, and thus do not contribute to volumetric changes in deposits due to hydration and dehydration. These salts primarily precipitate out of solutions, creating crystallization pressure within the pore spaces of deposits. In contrast, certain

perchlorates, like hydrous sodium perchlorate ($NaClO_4 \cdot 2H_2O$) and hydrous magnesium perchlorate ($Mg(ClO_4)_2 \cdot 6H_2O$), are capable of dehydrating to $NaClO_4$ and $Mg(ClO_4)_2$ and vice versa (Gough et al., 2011, https://doi.org/10.1016/j.epsl.2011.10.026). These minerals are likely capable of causing volumetric changes in deposits if they are present.

*ii. Among the references cited, only Dang et al. (2020) reported polygonal morphologies on Mars, with the interpretation involving desiccation. How can formation processes proposed in this study contribute to a better understanding of Martian polygons?*

Reply: As we stated in chapter 5.4 (line 636-638) most of the Martian patterned ground studies interpret that polygon formation on Mars is based on periglacial polygon formation (thermal contraction processes). We do not wish to claim that the observations of polygonal patterned grounds on Mars, attributed to ice, are incorrect. Instead, with our findings of similar structures in the Atacama Desert and the shared observations of high sulphate content on the Martian surface (based on remote satellite data), we aim to suggest that the proposed sulphate polygonal patterned ground process could also be occurring on Mars and should be considered in future research.

*d. Recommended references:*

*- Rapin et al., 2023, Nature, https://doi.org/10.1038/s41586-023-06220-3*

*This work reported Ca/Mg- sulfate-enriched polygonal ridges at Gale Crater, Mars.*

*- Cheng et al., 2021, Geomorph., https://doi.org/10.1016/j.geomorph.2021.107695*

*This study reported polygonal features controlled by Ca/Na-sulfates at the Qaidam Basin and discussed the implications for Martian polygons.*

*- The enrichment of hydrous sulfates (gypsum and aluminite) without halite in the surface crust at the studied site (Section 5.2 and line 510) is quite interesting to me. The presence and distribution of hydrous sulfates on the surface of Mars have intrigued the community (Gendrin et al., 2005, Science: https://doi.org/10.1126/science.1109087 and references therein). Extrapolating this study to Mars from this perspective may provide valuable insights. Nevertheless, this might deviate from the original focus. Thus, I leave it here for open discussion.*

*- (following the point above) Zhu et al., 2024, Geomorph., https://doi.org/10.1016/j.geomorph.2023.108934*

*This study reported halite enrichment on the surface of polygons in the arid environment in the Qaidam Basin. It is also intriguing to explore the factors that may cause such differences in mineral enrichment.*

Reply: Thank you very much for the recommended references. We have cited them in the introduction (line 135-137) and in the discussion (line 642) to reference additional studies on sulphates and polygon structures on Mars.

In line 592-594, we mentioned that the potential presence or absence of halite in the Aroma fan outcrop could be influenced by a sampling or outcrop bias, given that the outcrop had been exposed to atmospheric conditions for weeks or even months. Due to this uncertainty regarding the presence of halite, we opted not to discuss it in further detail.

**R(1): Minor comments:**

*Lines 43 & 103: The missing reference of Amundon (2018) in the bibliography list. Please also check other references.*

Reply: Apologies, we have added the missing reference in the bibliography list.

*Lines 57-59: Need some rewording.*

Reply: We have slightly reworded the sentence to improve readability. Line 81-85: *"The direct precipitation of anhydrite from a solution results in the formation of the calcium sulphate polymorph β-anhydrite (β-CaSO$_4$; insoluble), which is thermodynamically stable under the ambient conditions prevailing in the Atacama Desert (Tang et al., 2019; Beaugnon et al., 2020). As a consequence, this anhydrite polymorph occurs naturally in evaporite deposits (Beaugnon et al., 2020)."*

*Line 66: At the beginning of a paragraph, "such processes" refers to?*

Reply: It refers to the sum of all salt dynamics which are described in the paragraph before (swelling and shrinking and dissolution and reprecipitation of calcium sulphate). We have rephrased the line 93-95: *"The occurrence of salt-related processes, including swelling (hydration) and shrinking (dehydration), dissolution and precipitation in salt-bearing deposits within the hyperarid core of the Atacama Desert is linked to the persistence of hyperarid conditions since at least the Early Miocene (Dunai et al., 2005; Evenstar et al. 2009; Jordan et al., 2014; Evenstar et al., 2017; Ritter et al., 2018, Ritter et al. 2022)."*

*Lines 66-96: It seems like this part can be briefer in the introduction. Some can be moved to the section about the regional setting; other can be moved to the discussion regarding the moisture source/input.*

Reply: Thank you for your suggestion. We have decided to leave this part in the introduction, as we focus on the Aroma site in the regional setting rather than the general environment of the Atacama Desert, which we introduce in the introduction. We also think that moisture source/input is appropriate in the introduction, as the described salt dynamics require moisture

and need to be outlined in the introduction. We have referred to those moisture sources in the discussion section 5.4 (line 612-622).

*Line 103: Additional references should be included.*

Reply: We agree, we have added additional references. Line 131-133: *"The identification of wedge-forming processes in (hyper-)arid environments suggests that similar processes may also contribute to the formation of extra-terrestrial geomorphological features, such as those observed on Mars (e.g. Sager et al., 2021; Sager et al., 2023)."*

*Fig. 1A: Add the legend for the red squares or mention them in the captions. The sketch in Fig. 1E is excellent.*

Reply: We have mentioned the red squares in the figure captions.
Thank you!

*Lines 148-151: Fig. 2 seems more like results rather than serving as the illustration of the collected samples. Fig. 3 can be moved to Section 3 to illustrate the sampling locations.*

Reply: Apologies for the confusion and thanks for your suggestions. Fig. 2 ($^{21}$Ne exposure dating results) is indeed a results figure, we have corrected the figure reference in line 183. You are right, Fig. 3 also fits in section 3. We have moved the figure into section 3 where we describe the outcrop. We have also adapted the figure references in the text as the old Fig. 3 is now Fig. 2.

*Add in-text Fig. citations to enable easy navigation to photos of the samples.*

Reply: Thanks for your suggestion. Unfortunately, the in-text figure citations in my Word document do not function correctly when converting it to PDF.

*Line 234: Are these soil cracks related to desiccation? Please also clarify the terms: "soil" vs "salts" vs "samples", and "soil cracks" vs "salt wedges", throughout the text.*

Reply: Yes, the soil cracks are formed due to desiccation (dehydration of gypsum) in the sediment. We refer to our comment on the major concern of reviewer 2 → Soil crack formation and wedge cementation.

*Line 268, 273-276, and Fig. 4: It is recommended to directly use "wedge center" and "periphery" (and label them in Fig. 4B) instead of "LP" and "RP."*

Reply: Thank you for your suggestion. We have directly used "wedge centre" and "periphery" as recommended in line 311/314, 318/19. However, we have retained "RP" and "LP" in Fig.

4B because the XRD subsamples are labelled ARO18-08-RP-[Number] and -LP-[Number]. This allows readers to easily identify the samples based on the figure's description.

*Line 306 & 336: It would be more concise to mention "aluminite" specifically instead of using the phrase "evaporite content (except for Ca-sulfate).*
Reply: Thank you for your suggestion. We have deleted "evaporite content" and mentioned aluminite instead in line 379.

*Line 316: The dimension of the crack size should be specified.*
Reply: We have specified the dimension of the crack size in line 359.

*Fig. 5: Add a scale bar for the sample.*
Reply: There is already a scale bar (black and white bars) for the crust sample next to the ICP and XRD subsample points.

*Line 381: "Formed by subsurface pressure" is confusing.*
Reply: We have rephrased it and changed "subsurface pressure" to "desiccation processes" in line 456: *"The surface sediment is deposited in soil cracks formed by desiccation processes (predominantly dehydration of gypsum)."*

*Fig. 6: The red arrows only illustrate the swelling stress. How about the shrinking one?*
Reply: The shrinking stress is concentrated at the location of the soil crack. The red arrows represent the tensile stress caused by desiccation, while the other arrows, originating from the rhombuses, indicate the crystallization pressure that occurs during the precipitation of salts from solutions in the pore space of the alluvium. We have added arrows indicating the shrinking and swelling processes (white circle) in Fig. 6.:

[Figure]

*Fig. 7: Add the scale bar for all images. What materials, particularly the type of salts, are present in each site?*

Reply: Unfortunately, scale bars are not available for all images. Where possible, we have already included scales or provided scale ranges in the captions below the figures.

**Reviewer #2 (Christof Sager):**

*R(2): 1. An initial paragraph or section evaluating the overall quality of the preprint ("general comments")*

*Zinelabedin et al. (manuscript No.: egusphere-2024-592) present a compelling study on polygonal patterned ground featuring sulphate-cemented wedges overlain by a sulphate crust in the Atacama Desert. This research employs a robust multi-method approach to investigate these unique features, which stand out from previously studied polygon-wedge systems in the region. The authors attribute the formation of these wedges primarily to so-called haloturbation processes and the expansion-contraction dynamics (swell-shrinking) of calcium sulphate under hyper-arid conditions with sporadic rainfall events. The study is relevant for the readers of Earth Surface Dynamics as it makes a significant contribution to the research field of arid environments, while particularly enriching our understanding of patterned ground. Furthermore, its implications extend to extraterrestrial studies, particularly in understanding patterned grounds on Mars. Overall, the study is of good scientific quality and is well written, with the data presented appropriately. However, there are several aspects, including some*

*major concerns, that need to be clarified before the manuscript may be published. These concerns do not relate to the presented methods or acquired data, but rather to the interpretation of the results and the terminology used for salt-related processes and wedge formation. Addressing these issues should not pose significant challenges and will enhance the manuscript's clarity.*

Reply: We thank the reviewer for the constructive comments and the thorough assessment on our manuscript. We have considered most of the comments and suggestions, which have led to improvements in the manuscript. Below, we provide a detailed, point-by-point response to each of the reviewer's comments (in black italic font), along with explanations of the revisions made (in blue normal font and new edited parts in blue italic font).

*R(2): Major comments:*

*A major concern is the use of the term "haloturbation" as the dominant process for polygon-wedge formation, which I believe is not appropriate. First, this process is not explained in sufficient detail and is not clearly differentiated from other salt-related processes forming polygons/wedges, potentially leading to confusion for the reader. Second, the authors attribute the wedge formation in lines 413-415 mainly to shrinking processes due to phase transitions of sulfates, rather than directly to haloturbation, "which dominates in the polygon body, causing salt heave.". In my opinion, shrinking-swelling should not be seen equivalent to or be summarized under the term haloturbation (as in line 396), as shrinking-swelling is rather a term for volumetric changes in sediment or soil. Thus, shrinking process (or more general contraction) as the authors state in lines 413-415 is more appropriate than haloturbation as the main driving force for repeated soil cracking and thus, wedge formation. The authors already present very strong indication of hydration and dehydration processes by the presence of different calcium sulfate hydration forms with the XRD data. In contrast, haloturbation refers to the deformation of the original soil or sediment texture by the precipitation or dissolution of salts, but it remains too vague how it leads to meter-deep soil cracks. Therefore, the argument in line 381 that soil cracks are formed by subsurface pressure is questionable without further elaboration and contradicts the statement in lines 413-415 where contraction (shrinking) rather than expansion (which leads to subsurface pressure) is attributed to soil cracking.*

Reply: Thank you for your suggestions regarding the terminology of salt dynamics. We agree that the term "haloturbation" is not consistently or comprehensively addressed in the literature. In our view, haloturbation serves as a general term encompassing processes that disrupt or rearrange host deposits due to the dynamics and mechanisms of salt minerals. We propose that volumetric changes in the deposits, resulting from swelling and shrinking, along with the subsequent rearrangement of the alluvium (host deposit), should be encompassed by the term

"haloturbation", as these processes also contribute to the salt-related disturbance/deformation/rearrangement of host deposits.

We have added a general explanation in the manuscript to clarify why we use this term in line 69-78: *"The term "haloturbation" is typically used to describe all salt-related processes that modify the original structure of host deposits (rocks, sediments, soils). The presence of evaporites is a crucial factor in determining whether haloturbation is occurring. However, the definition of this term varies inconsistently in the literature. Some studies describe haloturbation as involving dissolution and reprecipitation (e.g., Rychliński et al., 2014), while others include swelling and shrinking due to the hydration and dehydration of salts (e.g., May et al., 2019).*

*We specifically use the term haloturbation to emphasize that salts are the primary agents and limiting factors in the processes encompassed by haloturbation (dissolution, reprecipitation, shrinking/dehydration, swelling/hydration). In this study, we focus on calcium sulphate-related haloturbation processes, as calcium sulphate is a dominant component of surface sediments in the hyperarid Atacama Desert (Ericksen, 1983; Rech et al., 2003; Ewing et al., 2006; Wang et al., 2014)."*

*For easier understanding, I suggest the following terms be clearly defined and differentiated in the introduction section: salt heave, clast/salt shattering as a form of salt weathering, contraction and expansion in the context of shrinking-swelling, thermal contraction, dehydration and hydration of minerals, and desiccation (as a crack formation mechanism). The term haloturbation is not necessarily needed when considering the above-mentioned concerns. It seems that these terms are not consistently defined throughout the literature, and their use and definitions vary across disciplines and studies which can lead to confusion for readers not familiar with these processes.*

Reply: We believe that desiccation, as a mechanism of crack formation at the described outcrop, should be considered as part of the overall shrinking process within the broader context of swelling and shrinking dynamics, rather than being treated as a separate term. As the wedges expand and the cracks widen over time due to the alternating swelling and shrinking processes, we chose to emphasize and clearly define the distinct processes of wedge formation at the outcrop. This approach helps avoid confusion stemming from inconsistent use of the term "haloturbation" in previous studies, while still acknowledging its general relevance. We thoroughly reviewed the terminology throughout the entire manuscript to ensure that all terms are precisely defined. We also added a paragraph, where we describe the issue with using the term haloturbation (we refer to our earlier comment and lines 69-78 in the manuscript).

*My understanding of the terms: contraction and expansion are more general terms used for volumetric changes in sediments and soils. Shrinking and swelling are commonly used for volume changes due to changes in water content in clay minerals with swelling potential, occurring in e.g., in playa environments where rain events lead to initial swelling of clays and subsequent drying/shrinking, resulting in desiccation crack polygons. For phase transitions between gypsum, bassanite, and anhydrite, the terms hydration and dehydration (as a form of desiccation) would be more appropriate, distinguishing them from the shrinking-swelling of clay minerals. Therefore, I suggest the dominant wedge formation processes be termed contraction due to dehydration of calcium sulfates rather than using the term haloturbation.*

Reply: We agree that the processes mentioned above should be defined using your suggested terms, such as "hydration and dehydration (as a form of desiccation)" and "contraction due to dehydration of calcium sulphates". Overall, we believe that these (sub-)processes occurring at the Aroma site should be classified under the term "haloturbation", as these subsurface dynamics are:

    a) primarily driven by salt minerals (distinguishing them from the clay-mineral processes mentioned earlier)

    b) responsible for disrupting the original alluvial fan deposits

We would also like to reference the study by Azam et al. (2000) (https://doi.org/10.1520/GTJ11060J), which evaluates the swelling effects resulting from the interaction between calcium sulphate phases (gypsum and anhydrite) with expansive clay. Based on this, we suggest that the terms "swelling and shrinking" are not exclusively linked to clays but can also be applied to calcium sulphate phases. These terms describe the process, but not necessarily the starting material. By using the term "haloturbation," we specifically indicate that calcium sulphate is the driving force behind these processes.

*A second concern is the interpretation of wedge cementation. Did the authors consider alternative explanations? The authors correctly state that in other studies, the wedges were largely free of salts or salt-poor. In Sager et al. (2023) (https://doi.org/10.1029/2022JG007328), a rain experiment was conducted showing that salt precipitation occurs on the surface of the wedges after wetting, likely due to upward movement of saline water along the wedges. Additionally, Sager et al. (2021) observed that the outer parts of the wedges had higher salt content than the inner parts. It was suggested that, over time and with sufficient rain events, salts migrate from salt-rich polygons towards the initially salt-poor wedges, eventually cementing them. Therefore, it is very interesting that Zinelabedin et al. observed so intensely cemented wedges. In Figure S.1 it appears that the wedge center shows a lower sulfate content (brownish colors) compared to the periphery (whitish color). The processes proposed by Sager et al. could also be relevant for the cementation of the Aroma fan wedges. Can the*

*authors discuss this possibility, or can it be excluded that the calcium sulfates in the wedges originated from the polygon? Addressing these alternative explanations would provide a more comprehensive understanding of the wedge cementation process and strengthen the study's conclusions.*

Reply: We have mentioned the studies of Sager et al. 2021 and 2023 and discussed their findings in regard to the Aroma fan wedge cementation in the discussion in line 420-442: *"The salt cementation of subsurface wedges in the Atacama Desert has also been discussed in previous studies. Sager et al. (2021) observed that the outer parts of sand wedges from the Yungay region exhibited higher salt concentrations relative to the inner parts. They suggested that, over time and with repeated rainfall, salts migrate from the salt-rich polygons to the initially salt-poor wedges, eventually causing their cementation. Furthermore, Sager et al. (2023) conducted a rain experiment, which demonstrated that salt precipitation occurs on the surface of the wedges after wetting. This is likely due to the upward movement of saline water along the wedges. Although we cannot completely rule out the possibility of post-formation cementation of the wedges, the cementation of the Aroma fan wedges may have resulted from calcium sulphate infiltration into the host sediments prior to the formation of the wedges, or alternatively directly from the surface soil. However, the available data do not allow a distinction to be made between these sources. As noted above, beta-anhydrite is both insoluble and stable under the current environmental conditions of the Atacama Desert. While some movement may occur from areas of high to low concentration, the mobility of calcium sulphate is significantly reduced once stable beta-anhydrite is formed.*

*Pfeiffer et al. (2021) conducted water infiltration in calcium sulphate-rich soils across different sites in the Atacama Desert. The study revealed a consistent sequence of soil horizons at all sites, characterised by a highly porous and conductive anhydrite layer above an impermeable, cemented gypsum layer. Significant water infiltration occurs mainly through the porous, conductive layer (such as the "chusca" layer at the Yungay site), while in the gypsum-cemented layer, infiltration is limited to vertical polygonal cracks, which are approx. 1.5 meters deep (as observed in the petrogypsic layer at Yungay). The processes of infiltration, dissolution, and reprecipitation of calcium sulphate at the Aroma fan site are thought to be concentrated around wedge structures, particularly within the cracks. The recent movement of fine particles, mainly through the cracks in the calcium sulphate-rich surface crust, as indicated by Pu isotopes, can be considered as a modern analogue of these processes."*

*2. Section addressing individual scientific questions/issues ("specific comments") and technical corrections*

***Title & Abstract:*** *The title and abstract should be changed regarding the major concern that haloturbation might not be the appropriate term. The summary would benefit from more*

*information, e.g., what is the composition of the polygons (the sediment/soil) between the wedges, how deep are the wedges and how many wedges were examined? The authors were able to determine an age for the crust. Since such ages are very important and rare, I would add them to the abstract.*

Reply: Lines 17-19: *" In contrast, it is hypothesised that haloturbation mechanisms, specifically the swelling and shrinking due to the hydration and dehydration of calcium sulphate, are the primary factors driving wedge formation at the Aroma fan site."* As stated in the abstract and previously discussed (see comments on major concerns above), we use the term haloturbation to encompass all potential processes that cause sediment/soil distortion due to evaporites. We rephrased/added information in line 12-14: *"This study aims to fill the existing knowledge gap by examining a network of vertically laminated, calcium sulphate-rich wedges that extend to depths of 1.5–2.0 meters in the alluvial subsurface of the Aroma fan in the northern Atacama Desert."*

Regarding the $^{21}$Ne exposure ages, we clarified in the manuscript that this age pertains to the surface quartz clasts of the Aroma surface (e.g. lines 249-251, 519) and is not an age for the surface crust, as chronological data for surface crust material and subsurface wedge material is lacking. With the exposure ages, our aim was to capture the time range marking the end of alluvial deposition and the onset of significant gypsum accumulation through atmospheric deposition (e.g. line 596/597). Plutonium data only indicate a downward migration of fine sediment over the last ~70 years (lines 539-540), instead an age of surface crust formation. To avoid misinterpretation (as exposure ages in this context have to be introduce and differentiated in more detail, as mentioned before), we chose not to emphasize the geochronological aspects in the abstract.

**Lines 11-12** *Consider rephrasing "post sedimentary features", since it remains questionable if wedges are completely post sedimentary features since they are formed by sediment deposition in the cracks and are accompanied by ongoing sedimentary processes as dry dust and salt deposition migration by percolating rainwater. Or rephrase, if with post sedimentary is meant after the deposition of the host material.*

Reply: We have used the term "post-sedimentary" to indicate that the formations occurred after the deposition of the local alluvium. To prevent any misunderstandings, we have deleted this term in line 11.

*14-16*

*From my point, these are very interesting polygon wedges, due to the high anhydrite content. However, wording suggests that the high anhydrite content lacking in other locations implies different formation process. However, it could be also post-formational processes that led to*

*these type of wedges. As stated above in the major concerns, I would add the possibility (maybe better in the discussion) that overprinting of sand wedges by calcium sulfate from the polygon could occur as a secondary process after initial wedge formation.*

Reply: Thanks for your suggestion. We have already discussed this in the comment above.

*16-17*

*Please rephrase under consideration of major concerns. Also, it is unclear who assume the process, the authors or the general scientific community. Better: "We assume contraction/shrinking due to dehydration of calcium sulfates to be the main driver wedge formation at the Aroma fan site."*

Reply: We have rephrased it in line 14-19: *"The subsurface wedges are characterised by their high anhydrite content, distinguishing them from the wedges and polygon structures found at other sites in the Atacama Desert. These structures appear to have been predominantly formed by thermal contraction or desiccation processes in playa-like environments. In contrast, it is hypothesised that haloturbation mechanisms, specifically the swelling and shrinking due to the hydration and dehydration of calcium sulphate, are the primary factors driving wedge formation at the Aroma fan site."*

Regarding the major concern, we refer to our comment above.

*23-24*

*Which are the other processes, do you include thermal contraction and desiccation as possible processes?*

Reply: Apologies for the confusion, we intended to only refer to the swelling and shrinking as well as dissolution and reprecipitation in the subsurface, which we summarized under the general term haloturbation. We have deleted the "other wedge-formation processes" in line 25, as we want to highlight the above-mentioned salt dynamics.

*Introduction*

*35-38*

*I would disagree that cryoturbation is the main mechanism for (non-sorted) polygon formation. Cracking and thus wedge formation is rather caused by stresses from rapid cooling or low temperature (thermal contraction), see Lachenbruch (1962) (https://doi.org/10.1130/SPE70). After initial cracking, seasonal freeze-thaw cycles allow water to enter the cracks and subsequently freeze, which forms wedges over time. However, sorted patterned ground, such as sorted stone circles (which do not have classical wedges as non-sorted patterns) are assumed to be a result of repeated frost heave processes (e.g., Kessler 2001,*

*https://doi.org/10.1029/2001JB000279). Thus cryoturbation is a process often occurring in the periglacial soils, but do not necessarily contribute directly to the pattern formation.*

Reply: We have rephrased it in line 37 and exchanged "cryoturbation" with "thermal contraction mechanisms". We have briefly explained sorted polygons in line 42-44.

**38-39** *This is correct, maybe the authors could elaborate when wedges are expected and when not. The differentiation between sorted and non-sorted patterns could help, as all non-sorted polygonal ground is defined by the presence of wedges (filled cracks) or unfilled cracks, that separate the polygons.*

Reply: We have briefly described sorted and non-sorted polygons in line 45-47.

**43-44** *Please rephrase the sentence, as "strongly differing" relates probably to the periglacial environments, but it reads as if this comparison is drawn to Mars, since it was discussed above.*

Reply: We mentioned in the end of the sentence that "strongly differing" is compared to the periglacial environment. However, we have rephrased it in line 52-53, to avoid confusion: *"In comparison to a water-rich periglacial environment, strongly differing environmental conditions prevail in the arid to hyperarid Atacama Desert, where landscape-modifying processes are influenced by severe water scarcity."*

**46**

*"wedge structures can also be found in the "Atacama Desert" instead of "here".*

Reply: We have replaced "here" with "Atacama Desert" in line 55.

**52**

*Buck et al. 2006 is very interesting but unfortunately only a conference abstract. Do the authors have found the appropriate manuscript to this study? Also, Buck et al. write:*

*"These features are interpreted to have formed through salt heave, which occurs when salt minerals cement soil grains creating the cohesion necessary for tensional stresses (caused by desiccation, and/or thermal contraction of salt minerals) to form contraction cracks. The contraction cracks are filled with eolian dust (salt/sediment), preventing their closure during periods of expansion caused by salt mineral precipitation and/or thermal expansion." Which means that wedges and cracks are formed by contraction rather than the salt heave itself. In Ewing et al. 2006 it is said: "The presence of sulfate polygonal prisms in multiple horizons (Fig. 10; Table 5c) suggests cyclical hydration and dehydration of gypsum/anhydrite (Chatterji and Jeffrey, 1963)" please consider putting this reference behind "dry environments" rather than "haloturbation". Or exchange haloturbation with contraction/dehydration.*

Reply: Unfortunately, we could not find any related study to the conference abstract of Buck et al. (2006). We have added the recommended reference of Chatterji and Jeffrey (1963) in line 62, but we prefer to keep the term haloturbation as we are combining several salt processes (see comment above).

Wedge formation at the Aroma site is not a product of salt heave. Salt heave processes are only assumed in the polygon body, which are responsible for the microtopographic signature (representing the patterned ground on the surface).

**60**

*What is was/where the other process or processes in addition to gypsification?*

Reply: Apologies for the confusion. We intended to explain that the precipitation of calcium sulphate in the pore spaces of the deposits leads to salt heave, as the salt precipitation exerts crystallisation pressure on the deposits. We mistakenly referred to this as a "another process […]" for swelling in the next sentence. We have rephrased the sentence in line 86: "The second mechanism, which we summarise under the general term haloturbation, is characterised by swelling and shrinking caused by phase transitions due to hydration and dehydration of calcium sulphates."

**66-67**

*Unclear wording, please rephrase. Do the authors want to say the dehydration /hydration processes were observed or potentially can be identified?*

Reply: We have rephrased it in line 93-95: *"The occurrence of salt-related processes, including swelling (hydration) and shrinking (dehydration), dissolution and precipitation in salt-bearing deposits within the hyperarid core of the Atacama Desert is linked to the persistence of hyperarid conditions since at least the Early Miocene […]".* We intended to point out that structures, indicative of these mechanisms, are observed in the Atacama Desert.

**2 Regional setting**

**126-128** *Regarding wedge depth, consider writing "between 1-2 m", since in Fig. 1 you state that cracks are all >1 m*

Reply: We have rephrased it in line 160: "~1–2 m".

**128-131**

*Consider adding these sentences to the results section rather than the regional setting section.*

Reply: Thank you for your suggestion. We have decided to leave this information in regional setting section (line 159-164), as this information supports Figure 1 which is also affiliated to the regional setting section.

**Figure 1**

Corrected Figure 1:

[Figure]

*General: Please homogenize the size of frames and labels, e.g., frame of panel C is smaller than E frame.*

Reply: As the individual figures do not have the same size, it is not appropriate to homogenize the size of frames and labels.

*Panel A: Please refer to the map of South America and the red dot in the lower left corner. Consider using a frame.*

Reply: We have briefly mentioned the map of South America in the Figure caption. For reasons of clarity in Figure 1, we have avoided to use a frame for this.

*Please give numbers for all isohyets, as it is missing for the one above 200 mm/yr.*

Reply: Apologies for that. We have added the missing number.

*Consider combining the symbols for wedge and polygon structures as except for dessication crack polygons, wedges and polygons are one system and it would simplify the map and enhance the position of published structures.*

Reply: Thank you for your suggestions. We chose to separate these symbols because some studies or sites exhibit only one feature (either patterned grounds or wedges), as is the case in our study, where only subsurface structures are observable in the field. By separating the symbols, it may provide better clarity for understanding other sites that display only one feature. For example, compared to the Aroma site, wedges at other locations might also be covered by a surface crust.

*There may be an unintended black line at the lower end of the 100mm/yr line and below "Gonzalez" further to the left.*

Reply: Thank you for making us aware of it. Normally it is a marking for the latitude and longitude, but it seems that it has been lost during file conversion from Word into PDF. We have corrected it.

*Is there a reason for the non-continuous coloring style of the DEM (orange vs gray, 2000m vs 2500 m)? If appropriate, consider using a continuous shading.*

Reply: Yes, we intended to highlight the Central Depression and Coastal Cordillera, as there occurs the majority of patterned ground and wedge structures (as well as our study site in the Central Depression). The region ranging from the Precordillera and higher is less significant to the subject of our study.

*PANEL C:*
*"Displaying a subsurface network of large soil cracks (>1 m depth) and vertically laminated wedge structures"*
*It could appear to the reader that soil cracks and wedges at the outcrop are two distinct features appearing next to each other, but in this case the belong together (I admit that in general cracks can be present without wedges and vice versa). Consider rewriting the sentence. e.g., "vertically laminated wedges with vertical soil cracks along their centers".*

Reply: We agree with your suggestion and have rephrased it in line 174-175.

*Panel D: If D is a close-up can the authors show the positions in C?*

Reply: No, D is not a close-up of a position within C. Panel D is just a close-up from another position, where the subsurface structures are very well recognizable and well developed to show them to the reader.

*Panel E:*

*This is a nicely drawn sketch. I wonder is there a reason for the more whitish colors in the upper part, and the more brownish colors in the lower part of the polygonbody, left of the '5'. The same for the surface crust on the right side of the sketch, which seems to be underlain by a darker surface crust? Please clarify.*

Reply: The colour differences indicate three-dimensional cuts of the polygons or polygon bodies within the outcrop. We have noted in the field that at some positions in the outcrop there are front and back cuts from adjacent polygons/polygon bodies. The majority of subsurface structures (which we presented in Figure 1D) are therefore polygon cross sections.

*142 „Detailed description of the wedge network is given in the result section" → I think this is not necessary and can be deleted.*

Reply: Thanks for your suggestion. We have deleted it in line 175.

**3 Material and methods**
**148**

*Consider using just 'trench wall' as in Fig. 1 instead of trench sidewall*

Reply: We have changed the term in line 182.

**148-151**

*It is unclear how many samples were collected and at which depths. E.g., What were the dimensions, weight and depth of the surface crust block? How many samples were taken below the crust, e.g., from wedges or polygon?*

Reply: We added the missing information in line 184-186. We have sampled two wedges (~40-50 cm depth) and two shattered clasts from the polygon body (one clast ~70–80 cm and ~100–110 cm depth). The surface crust thickness ranges from 0 cm to ~20 cm depth. Further information on the surface crust weight etc. is described in line 343-344, where the calculated density is mentioned.

**152**

*What kind of foil was used?*

Reply: We have used a cling film (line 188).

**154**

*The multi-methodological approach was only applied to the crust and wedges but not to the polygon? Was there a reason for this?*

Reply: The main focus of this study was the wedge evolution, which likely was influenced by the surface crust formation at the surface, as we discussed in the manuscript. Additionally, we have sampled two fractured clasts from the interior of the polygon body with anhydrite crusts at the clasts surface (see Figure S.3 in the Supplements). The anhydrite crusts and the anhydrite filling in the shatter veins from the clasts represent the (salt) material from the polygon clast body.

**159**

*Considering the high sulfate cementation, can the authors elaborate how was sampled, using a spoon, hammer, jackhammer?*

Reply: We used a hammer and chisel for wedge subsampling. We added this information in line 206.

**164**

*Please add the depth of the base of the crust.*

Reply: We have added this information in line 184. The depth was ~18-20 cm (thickness of the surface crust sample ARO18-02).

**165**

*Can the authors elaborate how the powder samples were generated? E.g.; hand grinded, grinding mill, grinding duration, temperature during grinding, drying of samples etc., also to ensure for the reader that a phase transition occurred accidentally during sample preparation.*

Reply: We have hand grinded the powder samples with an agate mortar to minimize phase transitions of calcium sulphate during sample preparation. We have added this information in line 217-218.

**167-169**

*If whole powder pattern fitting was used for 'Quantitative phase analysis', I recommend adding this information to the sentence.*

Reply: We have added this information in line 215.

**176**

*EDX stands for energy dispersive X-ray spectroscopy.*

Reply: We have corrected it in line 225.

*179-181*

*How did the authors verify that all the CaSO4 was removed from the sediment? In Figure 4, largest grain size is coarse sand, what happened with the fraction larger 2 mm, was it excluded? If so, I would add this information.*

Reply: We added in line 230-232: *"After treatment with the 10% NaCl solution, the clastic material was examined under a microscope to check for any remaining calcium sulphate in the sample or for coatings on the grains."*

The wedge ARO18-08 did not contain any grain size fraction larger than 2 mm.

*183*

*"Etched" suggests that not all carbonates were removed. I suggest to use another word for HCl treatment. Delete space between '10' and '%'.*

Reply: We have deleted the space between "10" and "%" in line 233 and exchanged "etched" with "was used" in line 233. There were hardly any carbonates present in the sample.

*185*

*It seems that ICP-OES is used here for the first time. When used for the first it should be spelled in full.*

Reply: We have added the full name in line 235.

*187*

*Please summarize the procedure by Voigt et al. shortly.*

Reply: We have briefly summarised the procedure by Voigt et al. in line 239-242: *"The procedure is based on 100 mg sediment, which was leached in deionized water (18.2 MΩ•cm) for 14 days at 25°C to extract the soluble salts from the samples. The concentrations of Ca and S were not considered for analysis as calcium sulphate phases were dissolved to saturation levels in the leachates."*

*200*

*How was sieved, wet or dry, by hand or sieving machine?*

Reply: We have dry sieved the samples by hand. We have added this information in line 253.

*209-211*

*Please rephrase to "we sampled dust/sediment from a cavity" instead of "we sampled a cavity".*

Reply: We have rephrased it in line 263: *"Afterwards, we sampled sediment from a cavity located […]".*

**4 Results**

*231-233*

*I would assume that the sulphate-cemented sediment contains also clasts smaller than pebbles. If the authors want to highlight that pebbles to boulder sized clasts are visible in a more fine-grained matrix, the sentence should be rephrased. If referring to 'Figure3 panel c', panel a and b must be mentioned first, or at least the whole figure 3.*

Reply: We have rephrased the sentence and the figure reference in line 285-287: *"Below the surface crust are clasts ranging in size from pebbles to boulders in a fine-grained calcium sulphate matrix ('polygon body'; see Fig. 2)."*

*233-234*

*From figure s3 it seems that ~30% of crack filling are other salts than anhydrite, can the others state which are these salts or add here the word "mainly" or "dominantly"? It is not clear if the authors refer to table S3 or Figure S3 → '(see S.3 in the supplementary material).'*

Reply: We have added the information in this part and corrected the figure reference in line 286-289: *"Many clasts are shattered, and cracks are mainly filled with calcium sulphate and 30.4 wt% aluminite in clast ARO18-04 and 17.6 wt% aluminite in clast ARO18-05 (see Fig. S.3 in the supplementary material)."*

*240 Figure 2*

*I think the combination of a photograph in the background with data presentation in the foreground is not necessary. Consider just presenting the data in figure 2, since the aroma fan surface is shown in Fig.1. alternatively add the oblique background image of the fan surface to Figure 1. If the whole photograph is directed into NE direction, I suggest to remove the NE notation in the top right corner as it is not appropriate.*

Reply: We agree that the image of the fan surface is not necessarily needed in Fig. 2 (now Fig. 3, as we changed the order of the figures, see comment above → reviewer 1). and have removed the background image of the fan surface.

Corrected Fig. 2 (now Fig. 3):

[Figure]

*Legend: 'Macroscopic' instead of 'macroskopic' Panel notation a) and b) is both at the Aroma site, why is it only notated at panel b? I suggest to remove 'Aroma site' at panel b or to add it at panel a as well.*

Reply: Apologies for the spelling mistake, we have corrected it.

Fig. 3 (now Fig. 2) We have added "Aroma site" to panel b) because the entire subsurface profile of the Aroma site is shown. Panel a) is only the detailed close-up of the surface crust. We have deleted "Aroma site" from the figure panel b) to avoid confusions and we have added this information to the figure caption in line 192 and 197.

Corrected Fig. 3 (now Fig. 2):

[Figure]

*The detailed investigation of the calcium sulfate polymorphs is very interesting. Can the authors elaborate how this polymorph of anhydrite was identified or differentiated from other polymorphs, e.g.; which database was used or provide diffractograms etc.?*

Reply: We have added the database Power Diffraction File-2 (PDF-2) in line 216/217.

*Please add the formula of halite as for the other evaporites.*

Reply: We have added the formula for halite (NaCl) in line 301.

***264-265***

*Please reference to the corresponding data or figure at the end of the sentence.*

Reply: We have added the reference in line 307-308: *"The (evaporite-free) clastic sediments in wedge ARO18-08 are dominated by sand, with medium and fine sand being the most abundant grain sizes (Fig. 4 B.3)."*

***265-267***

*I think this is very important observation, one could add, that this is visible by the color change of the wedges as well. The increasing cementation towards the periphery was also observed for the polygons in the Yungay region by Sager et al. 2021: "Furthermore, the cementation of the marginal and thus older parts of the SW indicates that rain events have occurred since the SW formation and the soluble content is not only leached downward but also migrated in horizontal directions. Although the vesicular horizon has formed in the margin of the SW, this cementation must take place without an intense deformation because the vertical lamination is still visible (Fig. 4)". However, the next sentence in **lines 267-269** is confusing as it states the opposite, if I understood correctly? Please clarify.*

Reply: Yes, we agree that the colour variations likely reflect different degrees of cementation within the wedge. However, both the wedge colours and SEM images show that, despite the general trend of higher cementation density towards the wedge periphery, the cementation density can also vary "randomly" within the laminations of the horizontal wedge transect. This is further demonstrated by a broader, whitish (vertical) layer near XRD subsample 16 (Fig. 4B) in wedge ARO18-08, which is close to but not directly at the periphery. Additionally, the area of the wedge around subsamples 17-20 appears to be more clastic-dominated compared to the whitish layer near subsample 16. The colour in the region of subsamples 17-20 (near the periphery) is also slightly darker than in the middle of LP, which can lead to confusion when correlating cementation with the colour of the wedge sediment.

We refer to our previous comment regarding the interpretation of wedge cementation.

***272***

*Instead of 'match' use 'correlate', and state, if possible, the $R^2$ value.*

Reply: As we did not intend to show a statistical correlation of the Ca and S plots, we have used the word "match" to just show the similarities between the two curves in line 315.

***281** Please add formula of aluminite.*

Reply: We have already provided the formula of aluminite in line 300, when we first mentioned the mineral aluminite.

***286 Figure 4***

*Panel B: The white dots indicate XRD sampling positions. In the supplementary material in Figure S.6, the XRD data is given for sample ARO18-02-001. Do I understand it correctly that this data corresponds to the most right white dote with the number 1. If not, please clarify. Panel B.1 it is hard to see, where the arrows for e.g., gypsum or anhydrite are pointing at. Could the authors highlight the clasts or part of the cement somehow, so that this becomes clearer?*

Reply: Apologies for the confusion, the subsample ARO18-02-001 is a subsample of the surface crust ARO18-02 (Fig. 5). The first subsample of the wedge ARO18-08 is "ARO18-08-RP1, which represents the number 1 in the Figure 4B. There is a typo in the figure caption of Fig. 4, it should be "ARO18-08" instead of "ARO18-02". We have corrected it in the manuscript in line 329. The black lines are meant to indicate the material in general (gypsum or anhydrite cement) rather than any specific "part" within it. Highlighting specific sections, as suggested, might lead to confusion by implying that we are focusing on particular structures rather than the cement as a whole. We believe the clasts are already clearly identifiable and do not require additional emphasis.

*Panel B.2: How were the wt% calculated, or is the data maybe semi-quantitative, then it should be [%]. Panel B.3 In the grain size diagram, it appears that a frame around the individual size fraction columns was used, leading to a poor recognizability (on my screen) of the small size fractions. If the fraction would be displayed without a frame, the visibility of the smaller size fractions could be increased. The authors also mention pebbles in cobbles in the sediment, why is the largest grain size fraction coarse sand? Was the other material excluded?*

Reply: The XRD data is quantitative, this is why we use wt%. We have added this information in the method section (XRD paragraph in line 215). The largest grain size fraction in the wedge is coarse sand, as the wedges show no larger grain sizes. We have also removed the frames in the diagram to increase the recognizability.

Corrected Fig. 4:

[Figure]

**297**

*"crust surface" instead of "crust top surface" as surface should be always 'top'.*

Reply: We have corrected it in line 340.

**304-306**

*Regarding the XRD data of crust surface, please provide sampling depth and refer to the corresponding data or figure at the end of the sentence.*

Reply: The crust surface has the sampling "depth" of 0 cm; therefore, we only wrote "crust surface" instead of "0 cm" in line 347-350. We have referred to the corresponding data in the end of the sentence in line 349 (→Table S.4).

**306-308** *'Subsamples taken from a few centimetres below the surface' → please provide sampling depth*

Reply: We have specified/added the sampling depths in line 349-350.

**312-314**

*Please refer to the corresponding XRD data or figure at the end of the sentence.*

Reply: We have added the reference in 358.

*316-317*

*Are the >15 cm the cracking width or depth?*

Reply: The crack depth. We have added this information in line 359 to avoid confusion.

**355 Figure 5**

*Figure 5 is a nice figure with a lot of information. But with the current organization, it can be overwhelming for the reader. I suggest reduce the presented information and organize the figure into panels with frames and/or labels. This will also increase readability of the figure caption. In Figure 4 the individual SEM images are not labelled, while here in figure 5 each image was labeled (SEM 1, SEM 2 etc.). In Figure 4 the grain size dots are grey, while in figure 5 they are orange. The XRD data is present without [wt%] but not in figure 4. Please use the same layout for both figures. The colored images right of SEM 3 are not explained in the figure caption, it appears that they belong to SEM 3, consider providing scale for these images as well. Also, SEM 3 appears two times, mark one as a zoom-in using frames, if this is the case. Please explain all abbreviations also in the figure caption, e.g.; GS = grainsize*

*The information of the 'general crust data' should be presented in the text instead of in the figure, to further clear the Figure.*

Reply: Thank you for your suggestions. We agree that Figure 5 could be overwhelming for the reader, so we have reduced the information for clarity. The figure has been divided into two parts: the most important data is now in the main manuscript, while additional data has been moved to the supplementary material. We have added "wt%" to the XRD data and renamed the second SEM3 as "SEM4." Additionally, we have clarified the coloured SEM images in the figure caption. As long as the figures remain internally consistent and maintain scientific quality, we have decided to keep the colouring of the individual analysis points as they are, since Figures 4 and 5 are separate figures and not directly related.

Corrected Fig. 5:

[Figure]

[Figure]

**Microscopic images of cement**

**Grain size**

*Discussion*

*General comments:*

*Please revise the discussion regarding the major comments (haloturbation vs dehydration of calcium sulfates/shrinking)*

*344-348 In the context of the major concerns, it remains unclear why the presence of Calcium sulfates in the wedges is an indication for haloturbation as dominate wedge formation process. As the authors nicely investigate different calcium sulfate phase and polymorphs it would be appropriate to state the mineral phases instead of the general term "calcium sulfate". As mentioned further above, cemented wedges were also observed in Yungay, but with a lower cementation degree, and a cementation of the wedges that corresponded to the cementation in the polygon, e.g., gypsum at 10 cm, anhydrite at 30 cm and halite at 70 cm depth. Consider adding a short paragraph comparing your results with wedge results from other studies in more detail, as this would be very interesting.*

Reply: We grouped gypsum and anhydrite under the term "calcium sulphate" because the wedges show varying proportions of these two minerals. For instance, wedge ARO17-03A is predominantly composed of gypsum, while wedge ARO18-08 is primarily dominated by anhydrite. We have added the XRD results of wedge ARO17-03A in Tab. S.4 in the

supplementary material. When discussing a specific wedge, we ensure to mention the exact mineral name. However, for broader statements regarding salt processes, we refer to both minerals collectively as "calcium sulphate."

We have incorporated the findings from Sager et al. 2021 and Sager et al. 2023 into the discussion on the wedge cementation in lines 420-442. We refer to our previous comment.

**352** *It is unclear what is meant with 'pattern' here?*

Reply: With "pattern" we meant that the solution supersaturation ratio is proportional to the crystallisation pressure.

**355-358** *Although the formation of anhydrite polymorph is still under debate, can the authors imagine or disprefer dehydration/shrinking processes that leads directly to β-anhydrite or would gypsum dehydration inevitable lead to γ-anhydrite that needs to be dissolved in a highly saline solution and the re-precipitated as β-anhydrite? In this sense it would be also interesting what anhydrite polymorph is dominating in the polygon. If one would assume that polygonal cracks form by dehydration, the polygon should contain γ-anhydrite (as a result of dehydration) and if the wedges are cemented over time by saline solutions, one could expect β-anhydrite. Another question would be if β-anhydrite can hydrate back to gypsum, do the author have references or hypothesis to this? β-anhydrite is often considered insoluble in the literature, do the authors know if this is also the case for geological timescales? Although it would speculative, the authors could give more insights on their thoughts regarding this topic.*

Reply: All of our XRD measurements detect β-anhydrite, including the anhydrite samples found within and on the fractured clasts from the polygon body. We have stated the β-anhydrite content of the wedges and the shattered clasts from the polygon body in line 405-407: *"The dissolution and precipitation processes of calcium sulphate are evident from the high content of the naturally occurring β-anhydrite in the wedge and in the shattered clasts from the Aroma fan outcrop (Fig. 1)."*

However, whether this formation occurs under typical desert conditions and the timescales involved remain subjects of ongoing debate (Ritterbach and Becker, 2020; Wehmann et al., 2023). Consequently, it is challenging to make definitive conclusions about the phase transitions occurring in the subsurface and their implications for the wedge-polygon formation. Because of the extremely rapid phase transition of gamma-anhydrite into bassanite, precise lab analysis is not feasible, as the transformation would already occur during sample transport. We also stated the possibility of the occurrence of γ-anhydrite in the soils of the Atacama in line 413-417: *"Shi et al. (2022) proposed that tunnels in the hexagonal crystal structure of γ-anhydrite from the Atacama Desert can incorporate cations of Si and P, which are thought to attenuate phase transition from γ-anhydrite to bassanite. The authors discussed that this*

*phenomenon enables γ-anhydrite to be prevalent in hyperarid environments such as the Atacama Desert and Mars."*

**358-361** *How would primary gypsum be differentiated from secondary gypsum?*

Reply: Primary gypsum is often represented by coarse-grained translucent or clear crystals, whereas secondary gypsum has rather a more fine and powdery structure of small crystals. However, these observations should be supported by lab techniques, e.g. thermogravimetric analysis in combination with XRD, (example of this technique combination in Seufert et al., 2009, https://doi:10.1016/j.cemconres.2009.06.018). Comparing the mass loss or water content between primary and secondary gypsum is interesting, as secondary gypsum may not fully rehydrate to the same extent as primary gypsum. However, this was not the scope of our study.

The gypsum in the Aroma fan samples is likely secondary, as it may have been "recycled" through atmospheric deposition and undergone multiple transformations during transport.

**362** "*silicon and phosphorous". Earlier in the manuscript, the authors use the element symbols (e.g., for S and Ca) instead of element names. Please homogenize.*

Reply: We have abbreviated the elements in line 415.

**364-367** *This is very interesting point, as it could explain dehydration (shrinking) processes for both anhydrite polymorphs γ-anhydrite and β-anhydrite.*

Reply: Yes, we agree that this is an interesting finding to interpret the shrinking mechanism likely based on the anhydrite polymorph transitions.

**368-415**

*Please edit this section regarding the main concern of haloturbation vs dehydration (shrinking). Also why does the saline solution exceeds saturation at greater depth and at which depth exactly?*

Reply: By "exceeding saturation," we mean that the saline solution has absorbed the maximum amount of salt possible as it moves deeper into the ground, dissolving additional salts along its way down, until reaching a certain depth where the salts begin to precipitate out of the solution. The exact depth at the Aroma outcrop is uncertain, but given that the polygons extend approx. 2 meters below the surface, we estimate this depth to be around 2 meters, or possibly even deeper.

**378-381** *This is a critical sentence that should be rephrased or deleted, as there is no reference or argument for crack formation by subsurface pressure, further it contradicts the shrinking*

*argument made by the authors in lines 413-415. The argument that expansion leads to upward deformation may be correct, but it is no explanation for soil cracking.*

Reply: We have revised the sentence and replaced "subsurface pressure" with "desiccation processes" in line 456.

**386-387** *Consider elaborating here about cementation of wedges by repeated rain events as mentioned above.*

Reply: Infrequent rain events, followed by the dissolution of surface evaporites and subsequent subsurface reprecipitation, may further strengthen the cementation of wedge structures. We refer to line 445-448 and to our comment above on the wedge cementation.

**387-391** *Consider adding that swelling shrinking applies probably mainly to the polygon. It could also be added that saline water migrates also through pore space and not only along the crack.*

Reply: We assume that the crack formation of the initial cracks is formed due to the shrinking of the polygon/cemented host material, but we believe that the cyclic shrinking of the calcium sulphate rich wedges mainly results into reopening of the cracks. We refer to our earlier comment on wedge cementation (line 420-442).

**392-394** *How do the authors know that salt heave processes intensify and do not occur e.g., with the same magnitude over time? I agree that salt heave mediated microtopography can visualize a polygonal surface pattern, but it is not the only reason why a pattern can be observed. In the Atacama Desert, the patterns are also visible due to color differences, e.g., darker or brighter wedges compared to the polygon body as a result of compositional differences or by morphology, e.g., high-center and low-center polygons. Morphology differences can be a result of erosion or soil deforming processes, including thermal expansion or salt heave (as the authors state correctly) leading to elevated polygon shoulders (low-center polygons). Therefore, please elaborate on the visualization of polygon patterns in the field.*

Reply: It is reasonable to assume that salt heave processes may intensify over time as the ongoing deposition and, particularly, the preservation of calcium sulphate in the system continues. This preservation is considered to be the primary driver of these processes. Since we did not observe any visible polygonal patterned grounds, our focus has been on the wedge formation. Due to the clear absence of patterned grounds at the Aroma site, we did not further examine the polygon morphology.

**403-406** *I agree that the outer parts of the wedges are likely the oldest part and I support the idea that salt-related processes may destroy initial lamination. But I disagree that wedge*

*formation is possible without initial soil cracking. Therefore, I suggest to elaborate on this or to remove the statement.*

Reply: We have removed the statement.

**408-410** *Do the authors mean phase transition from gypsum to anhydrite at the center or periphery, please clarify?*

Reply: The phase transition can take place across the entire horizontal transect of the wedge.

**410-415** *Again, I really support the attribution of crack opening to shrinking, but it should be differentiated from haloturbation.*

Reply: We refer to our earlier comment regarding the major concern about the use of the term haloturbation.

The initiation of cracks due to shrinking could potentially be influenced by the characteristics of the alluvium. However, there is no evidence to support this. Coarse alluvium, with sufficient pore space and low water content (due to good drainage), is unlikely to shrink significantly on its own.

*Figure 6*

*Figure 6 is a nicely drawn sketch, but considering my main concern, I suggest some re-editing. Panel 1: looks nice, no comments. Panel 2: I would show only the precipitation of salts and if wanted some haloturbation after the rain event (step 2). As a result of haloturbation also the original texture should change, leading to dislocation of some of the drawn clasts or some salt heave of clasts at the surface. I would move 'shrinking (dehydration)' and 'crack formation' and put it in the third panel (step 3), as this should occur after step 2. The red arrows that lead to the crack are due to contraction forces. However, this could be only visualized if the authors would show two wedges and a polygon in the center, where the arrows are directed towards the polygon center. As it is now in panel 2 shrinking and swelling seem to be directed in the same orientation. Also in Panel 3, the subsequent filling of the cracks by aeolian-derived sediment, leading to the first vertical lamina should be visualized. In Panel 4, I would show that the repetition of step 1-3 leads to wedges and clast shattering in the polygon over time. Then the formation of the surface crust can follow. If the authors do not recognize a microtopographic signature by salt heave I suggest to not draw it (or at least show it less intense).*

Reply: Thank you for your feedback on Fig. 6. We have taken your comments into account and have made some adjustments to the figure (see below) or provided responses to your suggestions:

Regarding your comment on Step 2: *"As a result of haloturbation also the original texture should change, leading to dislocation of some of the drawn clasts or some salt heave of clasts at the surface."*

➔ We have presented it this way because the pore space is believed to be significant, as indicated by this alluvial host material. In this scenario, evaporites would simply precipitate within the existing pore space, without necessarily altering or displacing the clasts within the body.

➔ We also included swelling and shrinking in Step 2 because, once salt cement is present in the pore space of the alluvium, these processes can occur at any time.

Step 3: *"Also in Panel 3, the subsequent filling of the cracks by aeolian-derived sediment, leading to the first vertical lamina should be visualized."*

➔ We have already visualized first vertical lamina of the wedge.

Step 4: *"In Panel 4, I would show that the repetition of step 1-3 leads to wedges and clast shattering in the polygon over time."*

➔ The repetition of Steps 1-3 (in Step 4) is depicted as the refilling and reopening, resulting from the processes shown in Step 3, that lead to the wedge structure seen in Step 5. We chose to separate these steps to maintain clarity in the figure.

We depicted the potential patterned ground formation with less intensity and added question marks, as we do not observe it on the surface but interpret that it may be present.

Corrected Fig. 6:

[Figure]

**437-476** *The section regarding the crust formwation is very interesting. Could the authors maybe elaborate on the extent of the crust. Is it only locally or covering the whole aroma fan, and is the crust also present in e.g., old gullies or river channels? How do the authors know the aroma fan surface is free of a surficial polygonal patterned ground, was there some reconnaissance by drone imaging or by satellite imaging (e.g., Google Earth)?*

Reply: The extent of the surface crust at the Aroma fan site is uncertain. Based on the literature (e.g., Evenstar et al., 2009), there is no patterned ground described on the Aroma fan surface. Therefore, we assume that the Aroma fan surface is largely free of patterned ground and is covered by a surface calcium sulphate crust.

Yes, we used Google satellite imagery to determine if any polygonal patterned grounds were visible near the sample site.

**480-482** *Please refer to the corresponding data or figure.*

Reply: We have added the data reference in line 559: "*[…] that could not be distinguished by ICP-OES results from ARO18-08) (see Table S.3 and S.4).*"

**488** *Aluminite is probably not detected in the clast, but rather in the filled fractures of the clasts right? "Aluminite is also detected in the fractured clasts in…" change to "Aluminite is also detected in the fractures of the clasts in…"*

Reply: We have changed it in line 567-570: "*Aluminite is also detected in the fracture fillings of the shattered clasts in the polygonal body of the Aroma fan outcrop[...]*"

**498-500** *Can the authors elaborate why "presence of sulphates reflects minor dissolution and reprecipitation of salts". Why only minor, please clarify.*

Reply: The term "minor" suggests that the process of dissolution and reprecipitation has occurred, but not to a significant degree. The numerous large cracks in the crust indicate that shrinking processes, likely caused by the dehydration of hydrous sulphates, have led to crack formation.

**504** *"as β-anhydrite also requires highly saline solutions to precipitate." Maybe the authors want to add "presumably" as they state earlier that formation of anhydrite polymorphs is still under debate.*

Reply: We agree and have rephrased the sentence in line 583-584: "*Given that β-anhydrite is thought to require highly saline solutions to precipitate, this is a plausible hypothesis.*"

**550-551** This seems like a completely new model of polygon-wedge formation by fog. Is this idea presented by Cereceda et al. or by Schween et al.? If so, the reference should be at the

end of the sentence. If it is a hypothesis of the authors, they should either explain it in more detail or consider removing it, as this would be an interesting topic but may not be within the scope of this study.

Reply: This idea is not presented in the two studies we cited. Instead, these studies were referenced to support the notion that the Rio Loa Canyon site is influenced by fog. We indicated that this region contains gypsum deposits and polygonal patterned grounds, suggesting that the formation of salt-induced wedge-polygon formation is likely encouraged by water derived from fog.

We chose to keep this section in the manuscript, because it provides a useful comparison or counterexample to the Aroma fan wedges, which were likely formed more by rain events than by fog. We have further specified this in line 631-633: *"However, the local influence of fog (e.g., Cereceda et al., 2008; Schween et al., 2020) and the presence of gypsum crusts (Mohren et al., 2020) suggest the potential for episodic salt-induced wedge-polygon formation in this region."*

**560** *Instead of haloturbation the authors could discuss the relevance of other salt-related processes for polygon formation on Mars (salt based thermal contraction/expansion or dehydration/hydration).*

Reply: We have replaced "haloturbation" with the above-mentioned salt processes in line 640-645: *"The presence of salt minerals on Mars (e.g. Clark and Van Hart, 1981; Osterloo et al., 2008; Hanley et al., 2012; Bishop, et al., 2014; Ehlmann and Edwards, 2014; Vaniman et al., 2018) and in particular hydrous sulphates (e.g. Gendrin et al., 2005; Dang et al., 2020; Rapin et al., 2023) suggests that salt-induced swelling and shrinking due to hydration and dehydration or thermal contraction could be additional potential mechanisms for the formation of polygonal patterned ground in regions with limited ground ice on Mars (less than 6% ground ice mass; Mangold, 2005)."*

**Figure 7***:*

*Figure 7 is a nice figure showing polygonal ground from different sites with different conditions, but I think the information content is too low to show it in the main manuscript, also since the comparison of polygonal ground between periglacial areas, Atacama Desert and Mars in the text is rather short, and that the polygons near Rio Loa were not investigated in this study. I suggest to remove the figure or to place it in the supplementary material.*

Reply: Thank you for your suggestion. We agree that the information content for this comparison is insufficient for the main manuscript. Rather than removing the image, we have moved the figure to the supplementary material.

***Conclusion***: *The conclusion can be shortened, as it reads more like a summary rather than providing new concluding remarks. E.g.; it could be highlighted that with this study another type of polygonal patterned ground has been identified and described in the Atacama Desert, which broadens the diversity of patterned grounds in the Atacama. Further, the analysis of the crust indicates that polygonal ground (usually a near surface feature) can be buried, thus enabling the possibility of multiple generations of polygonal patterned ground along a vertical ground profile. To make a link to Mars, the authors could mention the recent identification of buried polygonal ground underneath Utopia Planitia (https://doi.org/10.1038/s41550-023-02117-3). Given the new age constraints of the crust the authors could also highlight that polygonal patterned ground can be rather old/ancient landscape features in the study region (e.g., in contrast to recent gullies or fluvial deposits after the rain events from 2017).*

Reply: We have shortened the conclusion.

***845*** *The doi link seems not to be working for the reference.*

Reply: Unfortunately, the link does not work for the Word file, but if you type the link into the internet, it will work and take you to the corresponding study.

***Supplementary material***

*Figure S.1 Can the authors provide a scale for the photograph?*

Reply: We have added a scale in the figure.

*Figure S.2 Can the authors provide a rough size of the board used as scale for the photograph?*

Reply: We have added this information. The size of the board is ~24 cm.

*Figure S.3: Please add more information to the figure description, e.g., that XRD data is presented, consider using panels to organize the figure. Where are the two black lines pointing to? Is the XRD quantitative or semi-quantitative please add this information too.*

Reply: Thank you for your suggestions. We have already noted in the figure caption that these are "XRD samples," and "XRD" is also indicated within the diagram in the figure. The XRD results are quantitative, and we have added this information to the figure caption. The black lines indicate the subsample positions, as mentioned in the figure caption.

*Figure S.4: Do the authors maybe wanted to say "red points" instead of black points, as I don't see black points.*

Reply: Apologies for the mistake. We updated the figure but forgot to revise the caption accordingly. We have now corrected the figure caption.

*Table S.4: XRD results Can the authors state the unit for XRD analysis, e.g.; % or wt%? For the more exotic minerals such as Aluminite or Konyaite, could the authors provide individual diffractograms as done for Figure S.6. The authors could also highlight the main peaks for these minerals.*

Reply: We have updated the table caption to include the unit. Additionally, as you correctly noted, we have included a diffractogram for aluminite in Fig. S.6. Given that aluminite is more abundant than konyaite in all samples (as shown in Table S.4) and considering the generally higher concentration of aluminite in all samples, we believe that the discussion and interpretation should emphasize aluminite as the more significant mineral. In Figure S.6, the coloured bars below represent the characteristic "peaks" of each mineral or phase. The third row from the top (green) corresponds to aluminite. Given the large number of bars and to maintain clarity, we have chosen not to highlight every individual peak. We have explained this in the figure caption of Fig. S.6 in the supplementary material.

*Figure S.6: Can the authors add the information that the sample belong to the surface crust.*
Reply: We have added the information.

*Figure S.7: The illustration description reads redundantly, please shorten it.*
Reply: We have shortened it.

*Figure S.9: Could the authors leave a "white space" between the individual figures, as is the case with Figure S8?*
Reply: Since the right photograph in S.9 is a zoomed-in view of the left photograph, we prefer not to add white space between the images to emphasize that they belong together as a single set, unlike the separate images in S.8.

*Photogrammetry/Methology: What is meant with: "The sample was turned upside down and pictured in four runs" where it the four runs with similar camera orientation or was the camera angle changed for each run? If the picture count is stated for ARO18-02, it should be done for the other samples as well (or leave it out here, as it is stated in the Agisoft reports)*
Reply: The four runs maintained a consistent camera orientation, so the camera angle was not altered (ARO17-03A). The camera orientation was adjusted only for sample ARO18-02, as this specimen could not be rotated. We have removed the picture count for ARO18-02 as recommended.

*Why was the scaling of the 3D model done with distinct features of the sample when scale bars were laid out, or was it the combination of both, please state?*

Reply: Sample ARO18-02 was scaled using only distinct features due to the limitations of the equipment, specifically the turntable we used at the time, which did not provide sufficient space to place the scale bars alongside the sample.

*How was the density of ARO18-02 determined using photogrammetry if it could not be turned around?*

Reply: By adjusting the camera position, we captured the sample from a low angle, emphasizing the areas near the turntable. However, the bottom side of the sample could not be imaged and was instead interpolated. It is important to mention that the lower surface was relatively flat.

*For final publication, authors should consider presenting the Agisoft processing reports as a separate file, as page numbers, figure and table comments do not match the rest of the supplementary material, or editing the reports, which is likely to be more time-consuming*

Reply: Thank you for your suggestion. Unfortunately, we noticed during the file upload of our revised documents that only one document can be uploaded for the supplementary material. We have therefore combined the supplementary material with the Agisoft reports into one document. Since the Agisoft reports have a cover page, they are separated from the rest of the supplementary material, which should avoid the above-mentioned issues.